# Attention modulates neural representation to render reconstructions according to subjective appearance

Tomoyasu Horikawa [1][✉] & Yukiyasu Kamitani [1,2][✉]

Stimulus images can be reconstructed from visual cortical activity. However, our perception of stimuli is shaped by both stimulus-induced and top-down processes, and it is unclear whether and how reconstructions reflect top-down aspects of perception. Here, we investigate the effect of attention on reconstructions using fMRI activity measured while subjects attend to one of two superimposed images. A state-of-the-art method is used for image reconstruction, in which brain activity is translated (decoded) to deep neural network (DNN) features of hierarchical layers then to an image. Reconstructions resemble the attended rather than unattended images. They can be modeled by superimposed images with biased contrasts, comparable to the appearance during attention. Attentional modulations are found in a broad range of hierarchical visual representations and mirror the brain–DNN correspondence. Our results demonstrate that top-down attention counters stimulus-induced responses, modulating neural representations to render reconstructions in accordance with subjective appearance.

[1] Department of Neuroinformatics, ATR Computational Neuroscience Laboratories, Kyoto, Japan. [2] Graduate School of Informatics, Kyoto University, Kyoto, Japan. [✉]email: horikawa.t@gmail.com; kamitani@i.kyoto-u.ac.jp

Visual image reconstruction from brain activity produces images whose features are consistent with neural representations in the visual cortex given arbitrary visual instances[1–3], presumably reflecting the person's visual experience. Previous reconstruction studies have either examined how stimulus images are faithfully reconstructed or whether mentally imagined contents can be reconstructed without external stimuli. Meanwhile, although there is evidence that even stimulus perception is shaped by both bottom-up, stimulus-induced processes and top-down processes, whether images generated from brain activity during perception reflect the effect of top-down processes remains to be examined.

Among top-down processes, attention (or lack of attention) is known to affect visual experience[4–8] and brain activity[9–21] profoundly. Previous psychophysical experiments have shown that attention to visual stimuli can alter the appearance of stimuli, in which the perceived contrast of attended stimuli is enhanced[6–8]. Neuroscience studies have revealed that attention to specific visual features induces modulations of brain activity associated with the attended feature representations[11–21], which consequently enables decoding of attended information from brain activity patterns[14–18]. Although the link between the attentional modulation of perceptual contrast and that of neural activity was investigated using computational models[22], the study focused on single-feature stimuli and simple tasks based on single-neuron responses. It has been elusive how naturalistic visual features and their neural population-level representations, both of which are thought to have hierarchical organizations, are linked under the influence of top-down attention. Attentional modulations could impact images generated from brain activity to make them look similar to subjective visual experiences.

To address these questions, here, we used a state-of-the-art image reconstruction method (deep image reconstruction)[3] to reconstruct visual images from functional magnetic resonance imaging (fMRI) activity measured while subjects attended to one of two superimposed images with equally-weighted contrasts. Deep image reconstruction exploits the hierarchical correspondence between the brain and a deep neural network (DNN) to translate (decode) brain activity into DNN features of multiple layers[23,24], then create images that are consistent with the decoded DNN features[3]. Thus, using a deep image reconstruction model trained on fMRI responses to single natural images, we decoded brain activity during attention trials to produce reconstructions.

Using these methods, we first demonstrate examples of the reconstructed images under attention in comparison to those from single-image presentations. They show drastic alterations according to the target of attention given the same stimuli. The overall perceptual quality of the reconstructions is assessed by human raters, which indicates the resemblance of reconstructions to the attended images rather than unattended images. We then attempt to model the reconstructions with superpositions of attended and unattended images with variable contrasts. The results show that the reconstructions can be explained by superpositions with contrasts more weighted to the attended images, which are comparable to the appearance of equally-weighted stimuli under attention measured in a separate session. The in-depth analysis of the decoded DNN features that underlie the reconstructions reveals that attentional modulations are found in a broad range of hierarchical representations constrained by the correspondence between brain areas and DNN layers. These results illustrate that top-down attention counters bottom-up stimulus representations and modulates visual cortical representations to render reconstructions according to subjective appearance. The reconstructions appear to reflect the content of visual experience and volitional control.

## Results

### Visual image reconstruction from brain activity during attention

We collected fMRI data from seven subjects (initially five [Subject 1–5], then two more [Subject 6 and 7] at the editor's request during the revision, see Methods: "Subjects" for details) in two types of experimental sessions. In the training session, in which the data for model training were collected, subjects passively viewed presented natural images while fixating the center of the images (6000 trials; we reused a subset of previously published data of the reconstruction study[3] for the training session data of Subject 1–3; see Methods: "Subjects" and "Experimental design"). The test session, in which the data for testing the trained models were collected (all data of the test session were newly collected for this study), consisted of single-image trials and attention trials. In the single-image trials, subjects viewed presented images (ten unique images not included in the stimuli of the training session) as in the training session. In the attention trials, subjects viewed superpositions of two different images (all 45 pairs from the ten images). They were asked to attend to one of two superimposed images while ignoring the other such that the attended image was perceived more clearly (Fig. 1a). An entire test session of each subject contained eight attention trials for each pair and attention condition and 16 single-image trials for each single image.

We analyzed the fMRI data using the deep image reconstruction approach with which we previously demonstrated that perceived and imagined images were reliably reconstructed[3]. We first extracted DNN features using the VGG19 model[25] from the images presented in the training session. Linear regression models (decoders) were then trained to predict (decode) the individual DNN feature values from the patterns of fMRI voxel values in the visual cortex (VC) covering from V1 through the ventral object-responsive areas (see Methods: "Feature decoding analysis"). The trained decoders were then tested on the data from the test session (160 single-image trials and 720 attention trials). The decoded DNN features from each of the single-image and attention trials were processed with the optimization procedure to create a reconstructed image (Fig. 1b; see Methods: "Visual image reconstruction analysis")[3].

### Reconstructions from individual attention and single-image trials

Reconstructions from individual attention trials are shown in Fig. 2a (see Supplementary Fig. 1 for validations of decoders; see Supplementary Fig. 2 and Supplementary Movie 1 for more examples). The generated images appear to resemble the attended images; they tend to represent the shapes, colors, and finer patterns (e.g., faces) of the attended images to a greater degree than those of the unattended images. Notably, even for identical stimulus images, the appearance of reconstruction was strikingly different depending on the attention. The quality of successful reconstructions from attention trials appears to be comparable to the reconstruction quality from single-image trials (Fig. 2b, c).

We evaluated the reconstruction quality using behavioral ratings in a pair-wise identification task (see Methods: "Evaluation of reconstruction quality"). Human raters judged which of two candidates (attended and unattended images for attention trials; true [presented] and false images for single-image trials) is more similar to the reconstruction from each trial. Twenty raters performed the identification task for each reconstruction with a specific candidate pair (e.g., "post" and "leopard" for reconstruction with the target "post"). In the statistical analyses, the accuracy for attended image reconstructions was defined for each image pair by the ratio of correct identification across all trials and ratings. Since we did not control for the saliency of the images, each image pair might have had an intrinsic bias where one image became dominant over the other even without

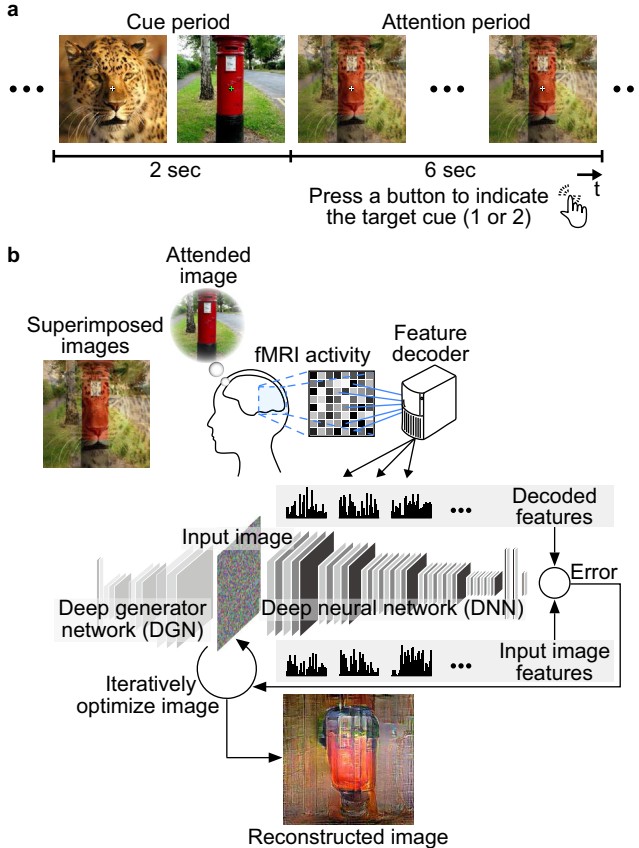

**Fig. 1 Overview of image reconstruction from brain activity during attention. a** Experimental design of attention trials. In each trial, two cue images and a superposition of the preceding two cue images (flashed at 2 Hz) were sequentially presented to subjects. During an attention period, subjects were asked to attend to one of two superimposed images indicated by green fixation color during cue periods while ignoring the other. Subjects pressed a button to indicate which of the first or second images were attended to for confirmation. **b** Reconstruction procedure. Given a set of decoded features for all DNN layers as a target of optimization, the method[3] optimizes pixel values of an input image so that the features computed from the input image become closer to the target features. A deep generator network (DGN)[45] was introduced to produce natural-looking images, in which optimization was performed at the input space of the DGN (see Methods: "Visual image reconstruction analysis").

top-down attention. Therefore, we pooled over the two attention conditions (attending to one or to the other) to cancel the potential effect of image saliency: if identification solely depends on the relative saliency of the two-component images regardless of top-down attention, the pooled identification ratios would cancel to the chance level. This yields a total of 45 data points in each subject that correspond to all the image pairs. To enable comparison, the accuracy for single-image reconstructions was also defined for each of the 45 image pairs by the ratio of correct identification across all trials and the ratings where the pair were presented as options for identification (45 data points corresponding to the 45 image pairs). Statistical analyses and tests were primarily performed within each individual subject, while the results averaged across subjects (45 data points, each representing the accuracy averaged across subjects for each image pair) are presented for summary and visualization purposes.

For attended image reconstructions, the identification accuracies were significantly higher than chance in four of the five original subjects (Fig. 2c; one-sided *t*-test, $p < 0.01$ for Subject 1–4,

and $p = 0.17$ for Subject 5, Bonferroni correction by the number of subjects; effect size [Cohen's $d$] = 1.251, 1.139, 0.890, 0.588, and 0.140 for Subject 1–5, respectively) and in the two additional subjects for replication ($p < 0.01$, Bonferroni corrected; effect size [Cohen's $d$] = 0.946 and 1.847, respectively). For single image reconstructions, the identification accuracies were significantly higher than chance in all original and additional subjects (Fig. 2d; one-sided *t*-test, $p < 0.01$ for all subjects, Bonferroni correction by the number of subjects; effect size [Cohen's $d$] = 3.191, 3.894, 4.307, 4.713, 4.211, 7.415, and 5.389 for Subject 1–7, respectively). The overall accuracy levels for attended image reconstructions were modest (95% confidence interval [C.I.] of the mean of the subject-averaged accuracies for 45 pairs, [57.1, 59.3]; see Supplementary Fig. 2c, d for reconstructions from poorly identified pairs). However, the accuracies were positively correlated between attention and single-image reconstructions across pairs (Fig. 2e; correlation with subject-averaged accuracies, Pearson's $r = 0.696$; permutation test, $p < 0.01$; similar results with individual subjects' accuracies). This indicates that attentional modulation was more pronounced for the pairs of the images that were easy to reconstruct when presented alone. Note also that subjects who were generally better at single-image reconstruction did not necessarily achieve greater accuracies for attended image reconstructions (e.g., Subject 4). The variance across subjects may reflect individual differences in the capability to exert attention.

**Attentional modulation modeled by image contrast**. Attention is known to enhance the perceived contrast of stimuli[6–8,13]. We thus sought to model the reconstructed images by superimposed images with biased contrasts (Fig. 3a; see Methods: "Evaluation of attentional modulation by weighted image contrast"). We created superpositions with weighted contrasts ranging from 0/100% to 100/0% (attended vs. unattended) for each pair, where 50/50% corresponds to the contrasts used for the stimuli in the attention trials. These weighted superpositions were given to the DNN to obtain their stimulus feature representations of individual layers (19 layers), which were then compared with neural feature representations to see attentional biases in multiple levels of visual representations. For each DNN layer, Pearson correlations were calculated between the decoded feature pattern from each attention trial and a set of DNN feature patterns of the super-imposed images with different contrasts. The contrast weight that yielded the highest correlation was considered to indicate the degree of attentional modulation. We could use the DNN features calculated from the reconstructions instead of the decoded features, but they highly resembled and yielded similar results in this analysis. Thus, the decoded features can be seen as the stimulus features of the reconstructions, too.

Examples from reconstructions with relatively high rating accuracies are illustrated in Fig. 3b (decoded from VC). The decoded features generally showed correlation peaks at greater contrasts of the attended images. The estimated correlations often peaked at 100% (i.e., attended image), indicating that representations regulated by top-down voluntary attention can override those from external stimuli in these examples.

In statistical analyses, the peaks from individual reconstructions were averaged within or across subjects for each image pair to cancel potential effects of image saliency as in the previous analyses (resulting in 45 data points corresponding to the image pairs). Overall, the peak correlations were shifted toward attended images in most DNN layers except for some lower layers (Fig. 3c; averaged across subjects). The mean of the peaks across layers was 56.4% with 95% C.I. [55.1, 57.6]. In individual subjects, statistically significant shifts were observed in 15, 17, 16, 11, 0, 13,

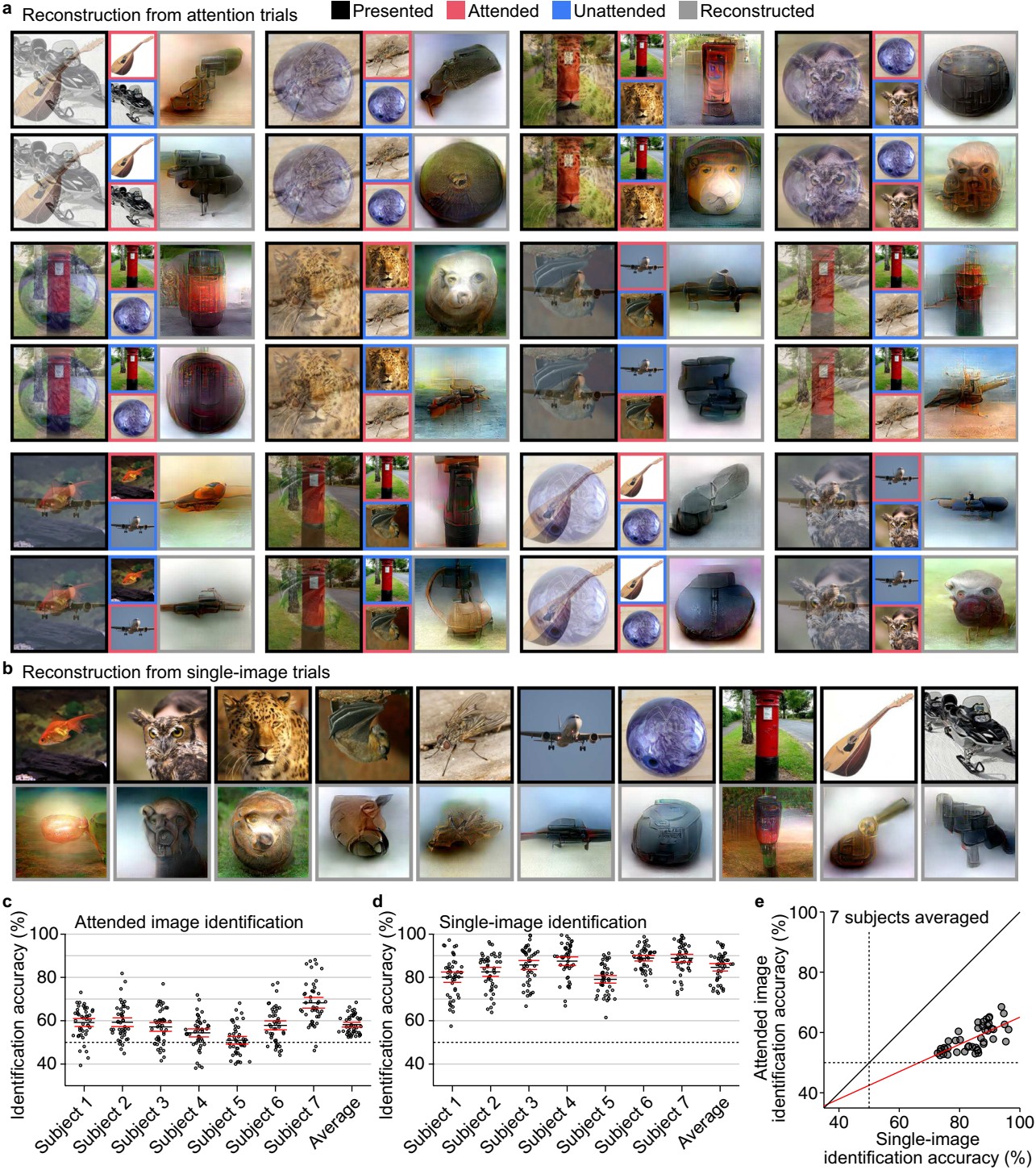

**Fig. 2 Reconstructions from individual attention and single-image trials. a** Reconstructions from attention trials. Reconstructions with relatively high rating accuracies are shown (see Supplementary Fig. 2 for more examples; see Methods: "Evaluation of reconstruction quality"). For each specific presented image, two reconstructions from the same subjects are shown for trials with different attention targets. **b** Reconstructions from single-image trials. Images with black and gray frames indicate presented and reconstructed images, respectively (see Supplementary Fig. 1c for more examples). **c** Identification accuracy based on behavioral evaluations for attended image reconstructions. Dots indicate mean accuracies of pair-wise identification evaluations averaged across samples for each paired comparison (chance level, 50%; see Methods: "Evaluation of reconstruction quality"). Black and red lines indicate mean and lower/upper bounds of 95% C.I. across pairs. **d** Identification accuracy based on behavioral evaluations for single-image reconstructions. Conventions are the same with Fig. 2c. **e** Scatter plot of attended and single-image identification accuracy based on behavioral evaluations. Dots indicate mean accuracies averaged over samples for each paired comparison and subject. The red line indicates the best linear fit.

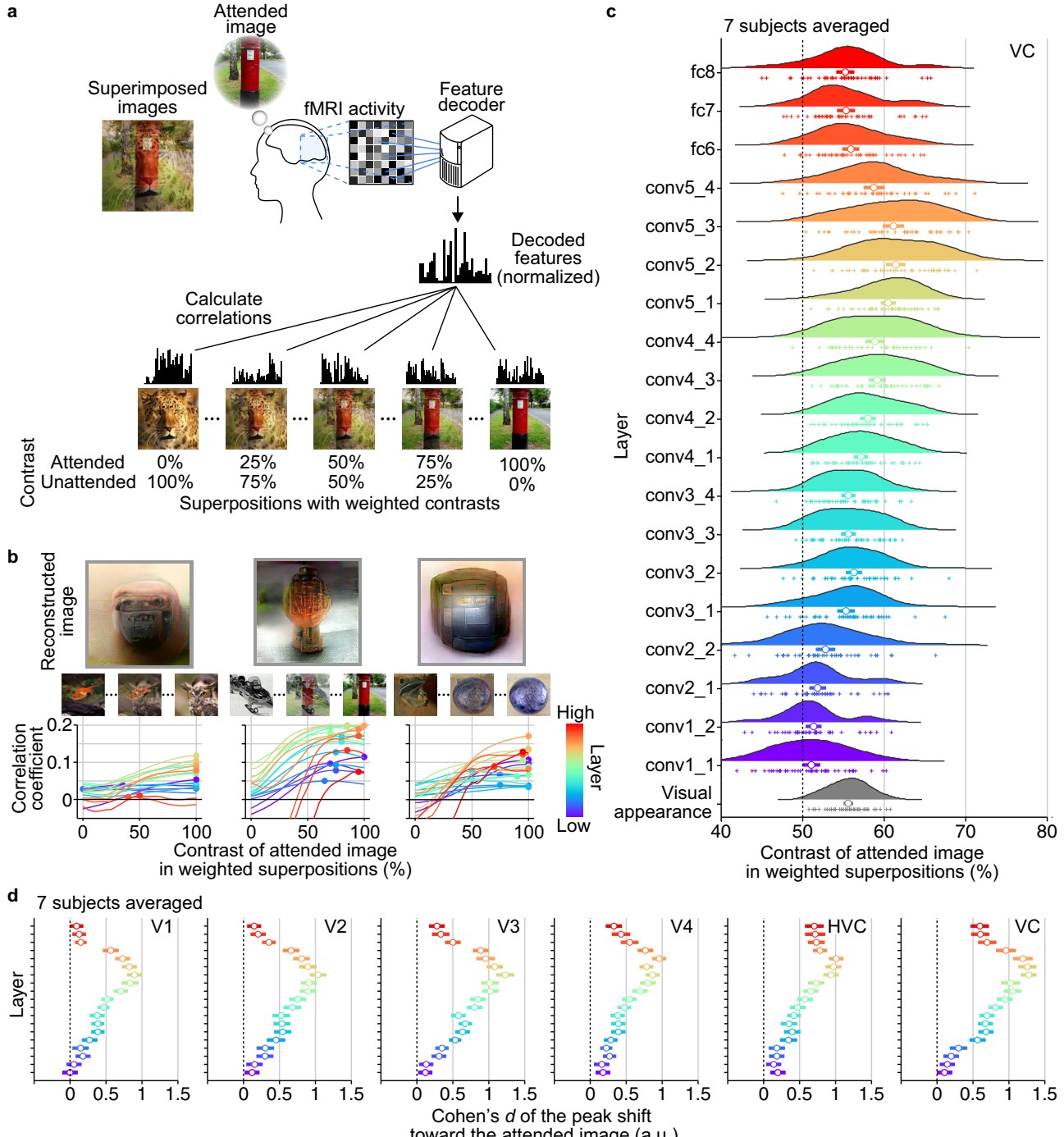

**Fig. 3 Attentional modulation modeled by image contrast. a** Evaluation procedure by weighted image contrasts. Correlations were calculated between decoded feature patterns and feature patterns computed from superpositions with weighted contrasts (5% steps; presented stimuli correspond to 50% contrast; normalized by the values in training images for each unit; see Methods: "Evaluation of attentional modulation by weighted image contrast"). **b** Correlation curves as a function of contrast for individual reconstructions with relatively high rating accuracies (decoded from VC). Circles indicate the contrasts with the highest correlations for each of 19 DNN layers denoted by colors. **c** Distributions of peak contrasts for each DNN layer (decoded from VC; averaged across subjects). The peak contrasts for 45 image pairs (cross markers) are shown with density plots (horizontal bars, 95% C.I.). The gray distribution at the bottom indicates contrasts of visual appearance evaluated in an independent behavioral experiment (averaged across subjects; see Methods: "Evaluation of visual appearance"). **d** Peak shifts at visual subareas (averaged across subjects). The effect sizes (Cohen's *d*) of the peak shifts (difference from 50%) are shown with 95% C.I. for each area and layer. See Supplementary Fig. 3 for the results of individual subjects.

and 17 layers out of 19 layers for Subject 1–7, respectively (Supplementary Fig. 3; one-sided *t*-test, *p* < 0.01, Bonferroni correction by the number of DNN layers). In an independent behavioral experiment, we measured the perceived contrasts of equally-weighted stimuli under attention by matching the

stimulus contrasts after the attention period. The matched contrasts (indicated by "visual appearance" in Fig. 3c) were comparable to the biases observed in the decoded features (55.6%, five subjects averaged; see Methods: "Evaluation of visual appearance").

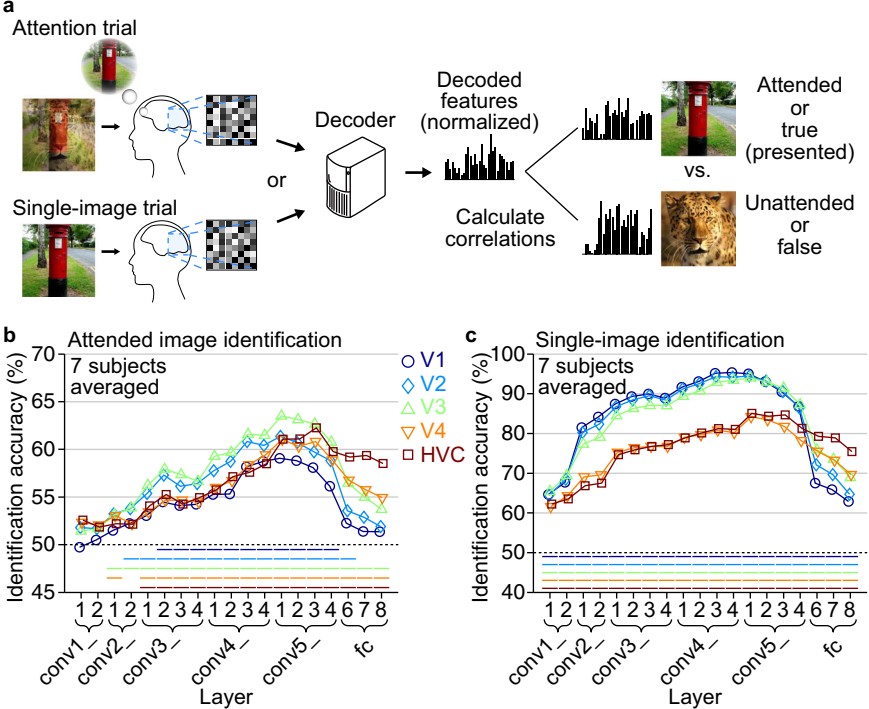

**Fig. 4 Identification by feature correlation. a** Identification procedure. Pair-wise identification of an attended image or a single presented image was performed via decoded features obtained from each trial (chance level, 50%; normalized by the values in training images for each unit; see Methods: "Identification analysis"). **b** Identification accuracy for attended image reconstructions (averaged across subjects). Mean identification accuracies are shown for all combinations of individual visual subareas and DNN layers. Statistical analyses were performed with the identification accuracies across 45 image pairs. Colored lines beneath data indicate the statistical significance of the difference from the 50% chance level (one-sided t-test, $p < 0.01$, Bonferroni correction by the numbers of brain areas and DNN layers), in which the power estimated by a post-hoc analysis was higher than 0.8 for all those significant results. **c** Identification accuracy for single-image reconstructions (averaged across subjects). Conventions are the same with Fig. 4b. See Supplementary Figs. 4, 5 for the results of individual subjects with effect sizes.

An additional analysis using five visual subareas (V1–V4 and higher visual cortex [HVC]) showed similar results with medium-to-large effect sizes mainly around middle DNN layers even at lower visual areas (Fig. 3d; shown in effect size). Similar results were observed in individual subjects (Supplementary Fig. 3). Significant peak shifts toward the attended image were observed in multiple combinations of brain areas and DNN layers for all subjects except Subject 5 (one-sided t-test, $p < 0.01$ in 70, 68, 65, 40, 0, 58, and 85 combinations from a total of 95 combinations [19 layers × 5 visual subareas] for Subject 1–7, respectively; Bonferroni correction by the number of combinations). These results indicate that robust attentional modulations are found across visual areas and the levels of hierarchical visual features as measured by the equivalence to biased stimulus contrasts.

**Identification by feature correlation.** We further investigated attentional modulations in terms of feature specificity in individual visual areas. Here, we performed a pair-wise identification analysis based on feature correlation, in which a decoded feature pattern was used to identify an image between two candidates by comparing the correlations to the image features (see Methods: "Identification analysis"). The value of each decoded feature was normalized by the mean and the standard deviation of the same unit's values in the 1200 natural images used for training feature decoders so that baseline differences among features were removed. The identification of attended images was performed for all combinations of areas and layers, and the results were compared with single-image identification (Fig. 4a). The identification accuracy was defined for each image pair by the ratio of correct identifications among all identifications where the pair

was used as options, yielding 45 data points for each area-layer combination and each of attended image and single-image identification.

The identification accuracies averaged across subjects are shown in Fig. 4b, c (only the means of 45 pairs are shown; see Supplementary Figs. 4, 5 for individual subjects' results). While V1–V3 showed markedly superior performance in single-image identification (Fig. 4c), especially at lower-to-middle DNN layers, such superiority is diminished in attended image identification (Fig. 4b). Regarding V1, attended image identification is generally poor across all DNN layers. Thus, V1–V3 appear to play a major role in representing stimuli, but not as much in attentional modulation. The attended image identification performances of different brain areas exhibit similar profiles, peaking at middle-to-higher DNN layers. The representations of these levels may be critical in attentional modulation.

A closer look reveals a hierarchical correspondence between brain areas and DNN layers. Attended image identification shows relatively higher accuracies from lower-to-middle areas (V2 and V3) with features of lower-to-middle layers (conv2–5) and from higher areas (V4 and HVC) with features of higher layers (fc6–8; Fig. 4b; see Supplementary Fig. 4 for individual subjects' results). This accuracy pattern generally mirrors the tendency found in the single-image identification performance (Fig. 4c) except V1. These results suggest that attentional modulation is also constrained by the hierarchical correspondence between brain areas and DNN layers for stimulus representation.

**Evaluation of amplitude modulations on decoded features.** To examine how attention affects the decoded features of individual

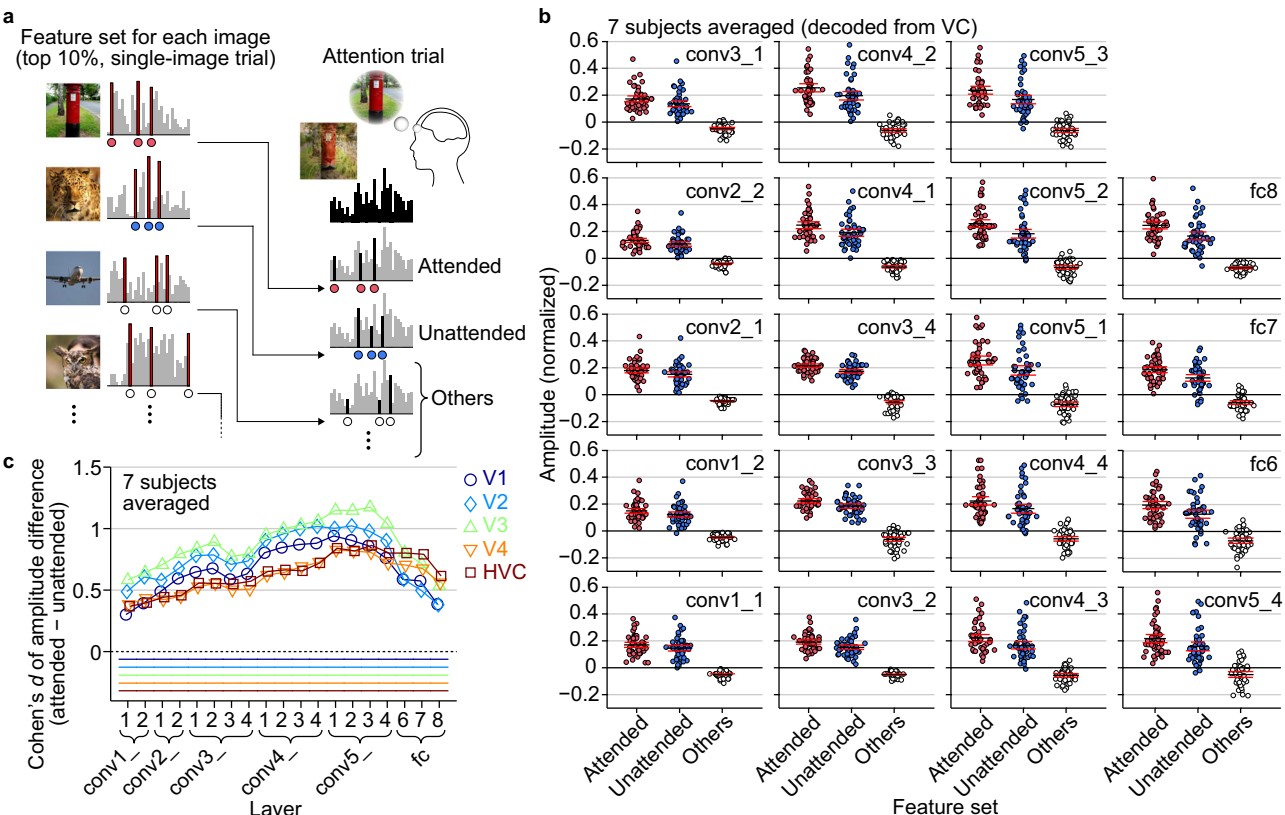

**Fig. 5 Evaluation of amplitude modulations on decoded features. a** Evaluation procedure by amplitude modulations. The amplitude of decoded features in each attention trial was evaluated for different sets of DNN features ("feature set" for the attended image, the unattended image, and the other images; normalized by the values in training images for each unit; see Methods: "Evaluation of amplitude modulations on decoded features"). **b** Amplitude of decoded features in attention trials (averaged across subjects). Mean amplitude averaged across units, trials, and subjects for the three types of feature sets are shown (decoded from VC; see Supplementary Fig. 6 for the results of individual subjects). Each dot indicates the mean amplitude of each image pair. Black and red lines indicate mean and lower/upper bounds of 95% C.I. across pairs. **c** Cohen's *d* of amplitude differences between the attended and the unattended feature sets at visual subareas (averaged across subjects). The effect sizes (Cohen's *d*) of the amplitude differences are shown for each area and layer. Statistical analyses were performed with the Cohen's *d* across 45 image pairs. Colored lines beneath data indicate the statistical significance of the difference (one-sided *t*-test, *p* < 0.01, Bonferroni correction by the numbers of brain areas and DNN layers). See Supplementary Fig. 7 for the results of individual subjects.

DNN units, we evaluated the amplitude modulation of the decoded features in attention trials by comparing the amplitudes of the decoded features of different sets of DNN units (see Methods: "Evaluation of amplitude modulations on decoded features"). For each combination of brain areas and layers, we first defined feature sets for each of the ten images used in single-image trials (also used as the component images in attention trials) by selecting the top 10% DNN units that exhibited the highest decoded feature values in single-image trials (Fig. 5a left; normalized by the values in training images for each unit and averaged across trials). For each attention trial, we compared the decoded feature values (normalized) between the feature sets for the attended image, the unattended image, and the other images (Fig. 5a right). The mean amplitudes for the three feature sets were averaged across trials for each image pair, yielding 45 data points for each combination of brain areas and layers.

The averaged amplitudes of the 45 image pairs for the feature sets are shown for each DNN layer in Fig. 5b (averaged across subjects, decoded from VC). Overall, the feature sets of attended and unattended images showed higher amplitudes than that of the other images, reflecting the fact that the presented images consisted of the attended and the unattended images (one-sided *t*-test, *p* < 0.01 for all layers, Bonferroni correction by the numbers of layers; see Supplementary Fig. 6 for the results of individual

subjects, in which *p* < 0.01 for all layers and subjects after Bonferroni correction by the number of combinations). Significantly greater amplitudes were found for the attended than the unattended feature sets in multiple DNN layers in the subject-averaged results (Fig. 5; all 19 layers; one-sided *t*-test, *p* < 0.01, Bonferroni correction by the numbers of layers). The results of individual subjects also showed significantly greater amplitudes in multiple layers for most subjects except Subject 5 (Supplementary Figs. 6; 9, 14, 17, 6, 0, 16, and 19 layers out of 19 layers for Subject 1–7, respectively; one-sided *t*-test, *p* < 0.01, Bonferroni correction by the numbers of layers and subjects). These results indicate that top-down attention is reflected in the amplitudes of a particular set of features relevant to the attended image. This may arise from amplitude modulations of neural feature representations because linear transformations link brain activity and decoded feature values. Interestingly, amplitudes for the feature set of the other images were slightly biased to negative values consistently across all layers, possibly indicating suppressive effects on irrelevant features during attention. Note that the value zero here is the mean feature value in each unit across many natural images (those used for decoder training). The stimulus feature values did not show such negative biases.

Additional analyses were performed for each combination of brain areas and layers, focusing on the difference between the

attended and the unattended feature sets (Fig. 5c; averaged across subjects; shown in effect sizes). Significant differences in mean amplitudes were found in multiple combinations of areas and layers. The results of individual subjects also showed significant differences in multiple combinations for most subjects except Subject 5 (Supplementary Figs. 7; 11, 32, 78, 12, 0, 51, and 92 combinations from a total of 95 combinations [19 layers × 5 visual subareas] for Subject 1–7, respectively; one-sided $t$-test, $p < 0.01$, Bonferroni correction by the number of combinations). The amplitude differences had medium-to-large effect sizes while showing the hierarchical correspondence between areas and layers consistent with the results in the identification analysis (c.f., Fig. 4 and Supplementary Figs. 4, 5). These results suggest a potential mechanism based on selective amplitude modulations of neural feature representations, which may underlie the effect of attention on reconstructed images.

## Discussion

This study investigated how top-down attention modulates the neural representation of visual stimuli and their reconstructions using the deep image reconstruction approach. Note that the decoding and reconstruction models had been trained with independent datasets of fMRI responses to single natural images, which did not involve the images used in the test (attention and single-image) trials. We found that the reconstructions from visual cortical activity during top-down, selective attention resembled the attended images rather than unattended images. While reconstruction quality varied across stimuli, successful reconstructions reproduced distinctive features of attended images (e.g., shapes and colors). When the reconstructions were modeled using superimposed images with biased contrasts, attentional biases were observed consistently across the visual cortical areas and the levels of hierarchical visual features. They were comparable to the subjective appearance of equally-weighted stimuli under attention. The identification analysis based on feature correlations revealed elevated attentional modulation for middle-to-higher DNN layers across the visual cortical areas. Attentional modulation exhibited a hierarchical correspondence between visual areas (except V1) and DNN layers, as found in stimulus representation. The feature amplitude analysis demonstrated the unit-level, selective modulations on features relevant to attended images. Our results demonstrate that top-down attention can render reconstruction in accordance with subjective experience by modulating a broad range of hierarchical visual representations.

We have shown robust attention-biased reconstructions, especially with image pairs whose individual images were well reconstructed when presented alone (Fig. 2e). The main results initially found from five subjects were replicated with two additional subjects collected during the revision (Fig. 2c). However, there were substantial performance differences across subjects. We found that subjects with higher performances in single-image reconstructions (e.g., Subject 4) did not necessarily exhibit better reconstructions of attended images (Fig. 2c, d) or greater attentional modulations (Supplementary Figs. 3–5). The difference in attentional modulation across subjects may be attributable to the individual difference in the capability to control attention. Exploring psychological and neuronal covariates with these differences may be an important research direction for future studies.

Reconstructions were explained by superimposed images with contrasts biased to the attended images, which were comparable to the appearance of stimulus images under attention (Fig. 3c). On average, decoded features were most strongly correlated with the stimulus features with biased contrasts of around 55/45%,

overriding the 50/50% contrasts in the stimuli. However, it should be noted that the peak biases were variable across DNN layers. Further, the visual features of biased stimulus images cannot account for the interaction of attentional modulations across layers. Thus, biased image contrast should be considered a rough approximation of attentional modulation in the visual system.

Another limitation of this study is the lack of explicit instructions to subjects regarding the strategy for directing attention to target images, which might partly explain the variations across subjects (Supplementary Fig. 4). Higher visual areas tended to be more closely linked to attentional modulation (Supplementary Fig. 3) potentially because subjects paid more attention to categorical aspects of the stimulus. Future experiments with explicit instructions regarding attention strategy would elucidate how specifically and flexibly attention can be deployed.

Because the identification analysis via decoded features was performed in a pair-wise manner using multivariate sets of features (Fig. 4), it is important to exclude potential confounding factors to correctly attribute the results to the effect of attention (as discussed in Naselaris & Kay, 2015[26]). Unlike conventional classification methods, our decoders were specifically tuned (trained) to predict the responses of individual DNN units (i.e., explicit models of representations as in Naselaris & Kay, 2015[26]), which could in part resolve the ambiguity of the source of the identification accuracies by attributing to the variations in feature representations. Our analysis of selected sets of features revealed that attention specifically enhanced the amplitudes of decoded feature values relevant to attended images (Fig. 5 and Supplementary Figs. 6, 7). Such selective amplitude modulations may make the pattern of decoded features similar to that of attended stimuli, resulting in accurate identifications.

While many previous studies of attention have reported effects of attentional modulation across multiple levels of the visual hierarchy in the brain, those studies have mainly focused on specific types of visual features and categories (e.g., edge orientations[14], motion directions[15], properties of receptive field models[19,20], and semantic categories[12,16–19]) using experimentally designed stimuli. In contrast, our approach is based on hierarchical DNN features that are discovered via the training with a massive dataset of natural images and thus are difficult for an experimenter to design. It enables us to examine millions of naturalistic visual features, many of which are relevant to neural representations in the human brain[24,27]. Attentional modulations were found in a broad range of hierarchical representations constrained by the correspondence between brain areas and DNN layers. Furthermore, the image reconstruction from decoded features enables in-depth examination of the extent and specificity of attentional effects. Admittedly, as this approach primarily relies on the validity of DNNs as computational models of the neural representation[28,29], the use of even more brain-like DNNs[30,31] may be needed to enhance the efficacy further to reveal fine-grained contents of attentionally modulated visual experience.

The present study is closely related to previous studies that have examined the relationship between a visual experience (appearance) and neural responses[13,22]. Störmer et al. (2009)[13] studied human EEG responses evoked by Gabor stimuli while spatial attention (left vs. right visual fields) was controlled by cross-modal cueing. The enhancement of apparent contrast of the cued stimulus was accompanied by an enlarged EEG response in the contralateral hemisphere. While this study found a link between appearance and neural activity under attentional modulation, our study extends it to fine-grained and hierarchical neural representations of complex and naturalistic stimuli, enabling the reconstruction of subjective percepts.

Cutrone et al. (2014)[22] investigated the effect of attention on appearance using simulated activities from simple contrast-response functions. They showed that the change in input baselines could account for psychophysical results on perceived contrasts of Gabor stimuli (gratings within apertures) under attention. Our findings are based on descriptive models that map brain activity to hierarchical image features of a fixed DNN model (and then to an image), which do not implement attentional modulation mechanisms. The comparison of the decoded feature values for the different sets of units (Fig. 5) suggests a mechanism mediated by amplitude modulations. It is consistent with the candidate models in Cutrone et al. (2014)[22] but does not distinguish between them. It also remains to be seen what mechanisms could underlie attentional modulation in the hierarchical representations of naturalistic stimuli. Our results from multiple layers of image and neural representations might provide constraints for potential models.

In the present study, we obtained image reconstructions from single trial fMRI activity (though statistical analysis was performed on pooled data). Despite the relatively low signal-to-noise ratio, the reconstructions were of comparable quality to those from trial-averaged data[3]. The single trial-based reconstruction of subjective images opens new possibilities of applications. It can be applied to new experimental designs that use real-time decoding/reconstruction and the feedback of the information. Furthermore, as the reconstruction reflects the content of experience and volitional control, it may provide a new means to express and communicate internal messages in the form of visual images.

## Methods

**Subjects**. Seven healthy subjects with normal or corrected-to-normal vision participated in our experiments: Subject 1 (male, age 34–36), Subject 2 (male, age 23–24), Subject 3 (female, age 23–24), Subject 4 (male, age 22–23), Subject 5 (male, age 27–29), Subject 6 (female, age 27–28), and Subject 7 (male, age 30–31). The first three subjects (Subject 1–3) were the same as those in a previous study (Shen et al., 2019). For these subjects, we reused a subset of previously published data (data for the training session, which was originally referred to as "training natural image session" of the "image presentation experiment"; available from https://openneuro.org/datasets/ds001506/versions/1.3.1), while newly collecting additional data (data for the test session). For the last four subjects (Subject 4–7), we newly collected a whole dataset (data for the training session and the test session). The sample size was first chosen on the basis of previous fMRI studies with similar experimental designs[3,23] ($n = 5$), and then we further collected data from additional two subjects (Subject 6 and 7) following the request by the editor through the revision. All subjects provided written informed consent for participation in the experiments, and the study protocol was approved by the Ethics Committee of ATR.

**Stimuli**. The stimuli consisted of natural color images, which were used in previous studies[3,23] and were originally collected from an online image database ImageNet (2011, fall release)[32]. The images were cropped to the center and resized to $500 \times 500$ pixels.

**Experimental design**. We conducted two types of experimental sessions: a training session and a test session. All stimuli were rear-projected onto a screen in the fMRI scanner bore using a luminance-calibrated liquid crystal display projector. The stimulus images were presented at the center of the display with a central fixation spot and were flashed at 2 Hz ($12 \times 12$ and $0.3 \times 0.3$ degrees of visual angle for the visual images and fixation spot, respectively). To minimize head movements during fMRI scanning, subjects were required to fix their heads using a custom-molded bite-bar and/or a personalized headcase (https://caseforge.co/) individually made for each subject except for the case where subjects were reluctant to use those apparatuses (a subset of sessions with Subject 5). Data from each subject were collected over multiple scanning sessions spanning approximately 2 years. On each experimental day, one consecutive session was conducted for a maximum of 2 h. Subjects were given adequate time for rest between runs (every 7–10 min) and were allowed to take a break or stop the experiment at any time.

**Training session**. The training session consisted of 24 separate runs. Each run comprised 55 trials that consisted of 50 trials with different images and five randomly interspersed repetition trials where the same image as in the previous trial was presented (7 min 58 s for each run). Each trial was 8 s long with no rest period

between trials. The color of the fixation spot changed from white to red for 0.5 s before each trial began to indicate the onset of the trial. Additional 32- and 6-s rest periods were added to the beginning and end of each run, respectively. Subjects were requested to maintain steady fixation throughout each run and performed a one-back repetition detection task on the images, responding with a button press for each repeated image to ensure they maintained their attention on the presented images. In one set of training session, a total of 1200 images were presented only once. This set was repeated five times ($1200 \times 5 = 6000$ samples for training). The presentation order of the images was randomized across runs. This training session is identical to that conducted in the previous study[3] (referred to as "training natural image session" of the "image presentation experiment"). The data for the last four subjects (Subject 4–7) were newly collected, whereas the data for the first three subjects (Subject 1–3) were adopted from the data published by a previous study[3] (https://openneuro.org/datasets/ds001506/versions/1.3.1).

**Test session**. The test session consisted of 16 separate runs. Each run comprised 55 trials that consisted of ten single-image trials and 45 attention trials (7 min 58 s for each run). In each single-image trial, images were presented in the same manner as the training session. In each attention trial, subjects were presented with a sequence of images, each of which consisted of two successive cue images (2 s, 1 s for each cue) and spatially superimposed images of the two cue images (6 s), and were asked to attend to one image (indicated by green fixation shown with either of the two cue images) of a superposition of two images while ignoring the other such that the attended images are perceived more clearly. During the attention period, subjects were also required to press one of two buttons gripped by their right hand to answer whether they correctly recognized which of the first and second cue image should be attended (percentages of correct, error, and miss trials among a total of 720 attention trials; 99.4%, 0.6%, and 0% for Subject 1; 98.8%, 0.6%, and 0.7% for Subject 2; 97.4%, 0.8%, and 1.8% for Subject 3; 99.9%, 0 %, and 0.1% for Subject 4; 93.5%, 3.5%, and 3.1% for Subject 5; 98.6%, 0.8%, and 0.6% for Subject 6; 99.3%, 0.7%, and 0.0% for Subject 7). In the test session, we used ten out of 50 natural images that were used in the previous study[3] ("test natural image session" of the "image presentation experiment"; these images were not included in the stimuli of the training session). The ten images were used to create a total of 45 combinations of superimposed images, and all these 45 unique superimposed images, as well as ten unique single images, were presented in each run with randomized orders (a total of 55 unique images were presented in each run). For each pair of superimposed two images, the number of trials to be the target of attention was balanced between the two images in every two consecutive runs over an entire session consisting of 16 runs. In total, there were eight trials for each condition of an image pair and attention, and 16 trials for each single image.

**MRI acquisition**. fMRI data were collected using a 3.0-Tesla Siemens MAGNETOM Verio scanner located at the Kokoro Research Center, Kyoto University. An interleaved T2*-weighted gradient-echo echo-planar imaging (EPI) scan was performed to acquire functional images covering the entire brain (TR, 2000 ms; TE, 43 ms; flip angle, 80 deg; FOV, $192 \times 192$ mm; voxel size, $2 \times 2 \times 2$ mm; slice gap, 0 mm; number of slices, 76; multiband factor, 4). T1-weighted (T1w) magnetization-prepared rapid acquisition gradient-echo (MP-RAGE) fine-structural images of the entire head were also acquired (TR, 2250 ms; TE, 3.06 ms; TI, 900 ms; flip angle, 9 deg; FOV, $256 \times 256$ mm; voxel size, $1.0 \times 1.0 \times 1.0$ mm).

**MRI data preprocessing**. We performed the MRI data preprocessing through the pipeline provided by FMRIPREP (version 1.2.1)[33]. For functional data of each run, first, a BOLD reference image was generated using a custom methodology of FMRIPREP. Using the generated BOLD reference, data were motion corrected using mcflirt from FSL (version 5.0.9)[34] and then slice time corrected using 3dTshift from AFNI (version 16.2.07)[35]. This was followed by co-registration to the corresponding T1w image using boundary-based registration implemented by bbregister from FreeSurfer (version 6.0.1)[36]. The coregistered BOLD time-series were then resampled onto their original space ($2 \times 2 \times 2$ mm voxels) using antsApplyTransforms from ANTs (version 2.1.0)[37] using Lanczos interpolation.

Using the preprocessed BOLD signals, data samples were created by first regressing out nuisance parameters from each voxel amplitude for each run, including a constant baseline, a linear trend, and temporal components proportional to the six motion parameters calculated during the motion correction procedure (three rotations and three translations). The data samples were temporally shifted by 4 s (2 volumes) to compensate for hemodynamic delays, were despised to reduce extreme values (beyond ± 3 SD for each run), and were then averaged within each 8-s trial (training session, four volumes), the last 6-s period of each trial (single-image trials in the test session, three volumes corresponding to second to fourth volumes in each trial), or 6 s attention period (attention trials in the test session, three volumes). For data from the test session, we discarded samples corresponding to error trials (miss or incorrect button responses) from the main analyses unless otherwise stated (e.g., Supplementary Fig. 2c, d; numbers of samples after removal, 716, 711, 701, 719, 673, 710, and 715 for Subject 1–7, respectively).

**Regions of interest**. V1, V2, V3, and V4 were delineated following the standard retinotopy experiment[38,39]. The lateral occipital complex (LOC), fusiform face area (FFA), and parahippocampal place area (PPA) were identified using conventional functional localizers[40–42]. A contiguous region covering the LOC, FFA, and PPA was manually delineated on the flattened cortical surfaces, and the region was defined as the higher visual cortex (HVC). Voxels overlapping with V1–V3 were excluded from the HVC. Voxels from V1–V4 and the HVC were combined to define the visual cortex (VC).

**Deep neural network features**. We used the Caffe implementation[43] of the VGG19 DNN model[25], which was pre-trained with images in ImageNet[32] to classify 1000 object categories (the pre-trained model is available from https://github.com/BVLC/caffe/wiki/Model-Zoo). The VGG19 model consisted of a total of sixteen convolutional layers and three fully connected layers. To compute outputs by the VGG19 model, all visual images were resized to 224 × 224 pixels and provided to the model. The outputs from the units in each of the 19 layers (immediately after convolutional or fully connected layers, before rectification) were treated as a vector in the following decoding and reconstruction analysis. The number of units in each of the 19 layers is as follows: conv1_1 and conv1_2, 3211264; conv2_1 and conv2_2, 1605632; conv3_1, conv3_2, conv3_3, and conv3_4, 802816; conv4_1, conv4_2, conv4_3, and conv4_4, 401408; conv5_1, conv5_2, conv5_3, and conv5_4, 100352; fc6 and fc7, 4096; and fc8, 1000.

**Feature decoding analysis**. We used a set of linear regression models to construct multivoxel decoders to decode a DNN feature pattern for a single presented image from a pattern of fMRI voxel values obtained in the training session (training dataset; samples from 6000 trials for each subject). The training dataset was used to train decoders to predict the values of individual units in feature patterns of all DNN layers (one decoder for one DNN unit). Decoders were trained using fMRI patterns in an entire visual cortex (VC) or individual visual subareas (V1–V4 and HVC), and voxels whose signal amplitudes showed the highest absolute correlation coefficients with feature values of a target DNN unit in the training data were provided to a decoder as inputs (with a maximum of 500 voxels).

The trained decoders were then applied to the fMRI data obtained in the test session (test dataset) to decode feature values of individual DNN units from fMRI samples constructed for each trial (samples from 160 single-image trials and 720 attention trials for each subject). The performance of the feature decoding was evaluated by calculating Pearson correlation coefficients between patterns of true and decoded feature values for each sample. To eliminate potential biases for calculating correlations due to baseline differences across units, feature values of individual units underwent z-score normalization using means and standard deviations of feature values of individual units estimated from the training data before calculating the correlations.

While similar decoding analyses were performed using a sparse linear regression algorithm in the previous studies[3,23], we here used the least-squares linear regression algorithm as the number of training samples (6000 samples) exceeded the input dimensions (500 voxels). We confirmed that results obtained from these algorithms were almost equivalent in decoding performance.

For the subsequent image reconstruction analysis, to compensate for possible differences in the distributions of true and decoded DNN feature values, the decoded feature values were normalized such that variances across units within individual channels/layers (groups of units within each channel for convolutional layers and all units within each layer for fully connected layers) matched with the mean-variance of DNN feature values computed from an independent set of 10,000 natural images. The feature values after this correction were then used as inputs to the reconstruction algorithm.

**Visual image reconstruction analysis**. We performed the image reconstruction analysis using a previously proposed method[3], which optimizes pixel values of an input image based on a set of target DNN features such that the DNN features computed from the input image become closer to the target DNN features. The algorithm was originally formalized to solve the optimization problem for reconstructing images from image feature representations, such as activations of DNN units in a specific layer, by inverting them to pixel values for a certain reference image[44]. Shen et al.[3] extended the algorithm to combine features from multiple DNN layers and to use DNN features decoded from the brain instead of those computed from a reference image. To produce natural-looking images, they further introduced a deep generator network (DGN)[45], which was pre-trained to generate natural images using the generative adversarial network (GAN) framework[46], and performed optimization at the input space of the DGN.

In this study, following the method developed by the previous study[3], we used decoded DNN features from multiple DNN layers (a total of 19 layers of the VGG19 model) and introduced the pre-trained DGN[47] (the model for fc7 available from https://github.com/dosovits/caffe-fr-chairs) to constrain reconstructed images to have natural image-like appearances. The optimization was performed using gradient descent with momentum algorithm[48] starting from zero-value vectors as the initial state in the latent space of the DGN (200 iterations; see Shen et al.[3] for details; code is available from https://github.com/KamitaniLab/DeepImageReconstruction).

**Evaluation of reconstruction quality**. We evaluated the quality of reconstructed images using behavioral ratings to quantify the similarity of reconstructions to attended (for attention trials) or presented single images (for single-image trials) via a crowdsourcing platform. In this behavioral rating experiment, human raters were asked to judge which of a pair of two candidate images (an attended image and an unattended image for attention trials; a true [presented] image and a false image for single-image trials) is more similar to a reconstructed image. The evaluation was conducted by 20 raters for each reconstruction with a specific candidate pair (e.g., "post" and "leopard" for a reconstruction with target "post"), and a ratio of correct identification of attended and presented images among all raters ($n = 20$) and candidate pairs ($n = 1$ for attention trials, $n = 9$ for single-image trials) were defined as the accuracy of a reconstructed image (this definition of the accuracy was used to categorize reconstructions into nicely or poorly reconstructed examples in Figs. 2, 3b, and Supplementary Fig. 2). The evaluation was conducted for all reconstructed images from samples of attention and single-image trials. In Fig. 2c, d, identification results of attended and presented images pooled overall trials ($n = 8$ for attention trials, $n = 16$ for single-image trials) and paired-images ($n = 2$, attended/unattended for attention trials and true/false for single-image trials) are shown.

**Evaluation of attentional modulation by weighted image contrast**. We evaluated the attentional modulation on decoded feature patterns (or reconstructed images) using superpositions of attended and unattended images with weighted image contrast. In this analysis, we assessed the effect of the attentional modulation by how the contrast showing the highest correlation with the decoded feature pattern deviated from the contrast of the presented image (50/50% for attended and unattended images, respectively). We first created superimposed images with weighted contrasts for every 5% steps ranging from 0/100% to 100/0% (attended vs. unattended) for each pair, in which 50/50% corresponds to the contrasts used for the stimuli in the attention trials (presented images). We then fed these weighted superpositions to the DNN (VGG19) to compute their stimulus features of individual DNN layers (19 layers). For each DNN layer, Pearson correlations were calculated between feature patterns decoded from individual brain areas and a set of DNN feature patterns of the superimposed images with different contrasts. The weighted contrast that yielded the highest correlation (peak) was considered to indicate the degree of attentional modulation (circles on lines in Fig. 3b). To cancel potential effects of image saliency, we then averaged peaks estimated from individual trials for each image pair (averaged across a total of 16 trials from two attention conditions except the error trials) to yield 45 data points that correspond to the image pairs and then used those data points to estimate the density, mean, and the confidence interval of the peak shifts for each DNN layer (Fig. 3c, averaged across subjects).

Additionally, the effect size of the peak shift was evaluated by calculating Cohen's d, in which the shifts of the peak position obtained from individual attention trials were first averaged for each pair (a total of 16 trials from two attention conditions except the error trials), and then the mean shifts toward the attended image relative to the chance (50%) were normalized by the standard deviation across pairs (Fig. 3d and Supplementary Fig. 3). The analysis was performed using features from individual DNN layers separately to see the degree of attentional bias at each level of visual representation and to avoid obscuring differences in effects from different layers due to substantial differences in the numbers of DNN units between layers.

**Evaluation of visual appearance**. To evaluate the visual appearance of stimulus images while paying attention to one of the overlapping images as in the fMRI test session (cf., Fig. 1a; see Methods: "Experimental design"), we conducted an out-of-scanner behavioral experiment with an available subset of the subjects who participated in the fMRI experiments (Subject 1, 4–7). Each trial in this experiment consisted of a cue period (2 s), an attention period (6 s), a white-noise period (0.1 s), and an evaluation period (no time constraint), in which the cue and attention periods were the same as those in an attention trial in the fMRI test session. During a white-noise period, we presented white-noise images (0.1 s, 60 Hz) at the same location as the presented images during the preceding cue and attention periods to diminish any potential effects of after-images. During an evaluation period, we presented a test image consisting of a mixture of two preceding cue images, which was initialized with a random contrast for the weighted superpositions. Subjects were required to change the stimulus contrast of the presented test image to be closer to the visual appearance of the image perceived during the preceding attention period by pressing buttons for control. After matching the contrast, subjects were allowed to start the next trial in 2 s after pressing another button for proceeding. The evaluation was performed for all 45 combinations of superimposed images and two attention conditions (a total of 90 conditions), which were separately evaluated in two separate runs with randomized orders (~15 min for each run). Each subject evaluated all conditions twice, and mean contrast averaged across all subjects ($n = 5$), repetitions, and attention conditions were used as a score for a specific pair (e.g., "owl" and "post"; a total of 45 data points in Fig. 3c).

**Identification analysis**. In the identification analysis based on feature correlations, correlation coefficients were calculated between a pattern of decoded features and patterns of image features computed from two candidate images (one for attended and the other for unattended images for attention trials; one for true [presented] and the other for false images for single-image trials). For each reconstructed image from attention trials, the pair-wise identification was performed with a pair of attended and unattended images (one pair for each sample). For each reconstructed image from single-image trials, the identification was performed for all pairs between one true (presented) image and the other nine false images that were used in the test session (nine pairs for each sample). To eliminate potential biases due to baseline differences across units, feature values of individual units underwent z-score normalization using means and standard deviations of feature values of individual units estimated from the training data before performing the identification. The image with a higher correlation coefficient was selected as the predicted image. In Fig. 4 and Supplementary Figs. 4, 5, the identification accuracy was defined for each image pair by the ratio of correct identification across all trials (except error trials) for attended image identification and by the ratio of correct identification where the pair was presented as options for single-image identification. These procedures yield 45 data points for each area-layer combination and for each attended image and single-image identification. In Fig. 4 and Supplementary Fig. 4, the accuracy is shown as a percentage. In Supplementary Fig. 5, the accuracy is shown as an effect size, which was calculated by first subtracting 50 (chance level) from the accuracy averaged for each pair (45 pairs) and then by normalizing the mean accuracies by the standard deviation across pairs.

**Evaluation of amplitude modulations on decoded features**. We evaluated the effect of attention on decoded features in attention trials by comparing mean amplitudes of decoded feature values between different sets of DNN features ("feature set"). For each combination of brain areas and layers, we first defined a feature set for each of ten images used in single-image trials (also used as the component images in attention trials) by selecting DNN units that exhibited the highest amplitudes in decoded features in single-image trials (top 10%, averaged across 16 trials). For each attention trial, we then used those feature sets to estimate mean amplitudes of decoded feature values averaged across DNN units that belong to each feature set of the attended image, the unattended image, and the other eight images. The estimated mean amplitudes for the feature sets of the eight "other" images were averaged to have a single scalar value for the "others" feature set. The same operation was performed for all attention trials, such that mean amplitudes for the three types of feature sets (attended, unattended, and others) were estimated for all trials according to the combination of presented image pairs. The trial-wise mean amplitudes were further averaged within each image pair (except error trials), yielding 45 data points for each combination of area and layer. The analysis was performed using normalized feature values, in which the baseline differences across units were removed in advance by applying z-score normalization to feature values of individual units using means and standard deviations of feature values of individual units estimated from the training data (1200 natural images).

The effect size of the amplitude difference between the feature sets of attended and unattended images was evaluated by calculating Cohen's $d$, in which the differences of mean amplitudes between the feature sets for attended and unattended images were first calculated for each of the 45 image pairs, averaged across pairs, and then normalized by the standard deviation across pairs (Fig. 5c and Supplementary Fig. 7).

**Statistics and reproducibility**. One-sided $t$-test was used to test the significance of the identification accuracies based on behavioral evaluations ($n = 45$; Fig. 2c, d), the significance of the feature decoding accuracies, and identification accuracies for the single-image trials ($n = 160$, decoded from VC, Supplementary Fig. 1a, b), the significance of the peak position shift from the baseline (50%; $n = 45$; Fig. 3b), the significance of the identification accuracies based on decoded DNN features ($n = 45$, Fig. 4b, c and Supplementary Fig. 4), and the significance of the difference of amplitude modulations between attended and unattended feature sets (Fig. 5c and Supplementary Figs. 6, 7). A Pearson's correlation coefficient between the identification accuracies of attention and single-image trials evaluated based on behavioral evaluations were tested by a two-sided exact permutation test ($n = 45$; Fig. 2e). A post-hoc power analysis was performed to estimate the power of the $t$-test ($n = 45$) for evaluations of identification accuracies via decoded feature patterns (Fig. 4 and Supplementary Fig. 4). The normality of the distributions analyzed with $t$-test was tested by Kolmogorov–Smirnov test.

Results from multiple subjects and trials can be considered as replications of the analysis. Using fMRI data of initially collected five tested subjects, the main findings were independently replicated from four subjects with multiple successful trials for each subject. Furthermore, at the request of the editor and reviewers during the revision, we have additionally collected data from two more subjects and confirmed the replicability of the main findings with those new subjects.

**Reporting Summary**. Further information on research design is available in the Nature Research Reporting Summary linked to this article.

## Data availability
The experimental data that support the findings of this study are available from open data repository (raw data in OpenNeuro: https://openneuro.org/datasets/ds001506/versions/1.3.1; https://openneuro.org/datasets/ds003430; preprocessed data and all source data underlying the graphs and charts presented in the main figures in figshare: https://doi.org/10.6084/m9.figshare.13474629).

## Code availability
The code that support the findings of this study are available from our repository (code for feature decoding: https://github.com/KamitaniLab/GenericObjectDecoding, https://doi.org/10.5281/zenodo.5722665 [49]; code written in MATLAB R2017b was used in this study; code for image reconstruction: https://github.com/KamitaniLab/DeepImageReconstruction, https://doi.org/10.5281/zenodo.5717775 [50], python 2.7).

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

## Acknowledgements
The authors thank Misato Tanaka for assistance with the reconstruction evaluation experiment, Mitsuaki Tsukamoto for assistance with data curation, and Benjamin Knight, MSc., from Edanz Group (https://en-author-services.edanz.com/ac) for editing a draft of this manuscript. This study was conducted using the MRI scanner and related facilities of Kokoro Research Center, Kyoto University. This research was supported by grants from the New Energy and Industrial Technology Development Organization (NEDO), JSPS KAKENHI Grant number JP15H05710, JP15H05920, JP17K12771, and 20H05705. JST CREST Grant Number JPMJCR18A5, and JST PRESTO Grant Number JPMJPR185B Japan.

## Author contributions
Conceptualization: T.H. and Y.K.; Methodology: T.H.; Validation: T.H.; Formal analysis: T.H.; Investigation: T.H.; Resources: T.H. and Y.K.; Writing—Original Draft: T.H.; Writing—Review & Editing: T.H. and Y.K.; Visualization: T.H.; Funding acquisition: T.H. and Y.K.

## Competing interests
The authors declare no competing interests.
