## [Transparent Peer Review File · Communications Biology]

Reviewers' comments:

Reviewer #1 (Remarks to the Author):

This paper demonstrates that it is possible to reconstruct images viewed by human subjects in an fMRI scanner in an ambiguous situation where two images are superimposed, and the subject is instructed to attend to one of them. Importantly, the reconstruction favors the attended image, demonstrating that the top-down modulation of neural signals by attention can be “read out” by fMRI decoding methods. This scientific conclusion, if not surprising (given prior knowledge from primate physiology), is important enough to merit publication. The reconstruction method and obtained images establish a new state-of-the-art for fMRI-based image reconstruction.

The paper is clearly written and structured, the arguments are sound and logical, and appropriate details are provided to allow replication of the results. In fact, the manuscript is in very good shape, possibly as a result of previous rounds of review in other journals (?). Furthermore, the decoding technique is noteworthy, drawing on state-of-the-art deep learning models. Finally, the novel dataset collected by the authors will be made available to the community on a data-sharing platform, and has the potential to become a standard dataset in the field. For all these reasons, I am enthusiastic about the manuscript, and I believe it can be published pretty much as it is.

Reviewer #2 (Remarks to the Author):

I thank the editor for the opportunity of reviewing this elegant study on the influence of top-down processes during the perception of stimuli. The topic is new, and it will add a significant contribution to the field of neural representations. However, few concerns must be addressed. A major issue concerns the effect size of the manuscript's results. As a general recommendation, I would ask to author to report the effect size and the power of their main findings.

1) In figure 3b, the top panel shows the correlation between the reconstructed image and the superimposed image. The graph shows that the peak correlation is around a low/medium effect size for most DNN layers for the first trial. Nevertheless, the correlations showed for the other two trials reach a small effect size only for few (highest) layers. These results indicate that only a few layers correlate with the weighted contrast. I would ask the authors to discuss how this result can be interpreted in terms of the degree of attentional modulation. I would also ask the author to statistically quantify the difference among the trials and modify the y axis to indicate the 0.1 correlation point (the least coefficient around which correlation can be assumed with a small effect size).

2) In figure 3b, the bottom panel represents the correlation between the reconstructed image and the superimposed image, averaged for trials and participants. The graph reveals that only a few layers reach the 0.1 coefficient. As above, I would ask the authors to discuss only the correlations that display a small to high effect size.

3) Similarly, the results displayed in Figure 3c seem to contrast with the results discussed in the text. In fact, the higher layers seem to display a weak correlation with the weighted superimposition only in higher visual cortices. This result appears to be confirmed by the identification accuracy analysis, where lower visual cortices exhibit lower performances in attended image identification.

4) Regarding the identification analysis, I would ask the authors to provide the power of their binomial and non-parametric analyses, given the low number of participants. Furthermore, the correction for multiple comparisons may be necessary for this analysis.

Minor points:

- In the results (Figure 2c; Supplementary Figure 4) and the discussion sections, the authors state that subjects with higher accuracy performance in single-image do not perform better at attended image reconstructions. I wonder whether the authors performed a within-group statistical comparison of single vs. attended accuracy performance to support this claim.
- Has the effect of the time gap (2 years) been controlled in the accuracy analysis?
- The links on page 17, line 380, and page 19 lines 447-450, redirect to "not found" content.

Reviewer #3 (Remarks to the Author):

Horikawa & Kamitani

This is an interesting manuscript that investigates how a neural network can decode and reconstruct images shown to people from recordings of their brain activity. The manuscript builds on earlier work by the same authors showing that such reconstructions from participants' brain activity can be similar to the image presented to the participant, to now examine how such reconstructions are affected by the exercise of voluntary attention. The principal finding is that the exercise of voluntary attention leads to changes in the reconstructed image, leading the authors to infer that this reflects a change in the brain activity that serves as input to the decoder.

This is a reasonable conclusion, but emphasises the challenge this paper has in terms of novelty. Much is known about how attention modulates brain activity, and broadly speaking this paper confirms what is already known. The novelty of this paper therefore rests primarily in the application of these decoding methods to brain activity rather than in any specific biological insights.

I have a number of more specific suggestions for improving the paper:

1. Tone down claims in the Introduction. Lines 29-31 are in my view overstated – there are quite a few papers showing that neural representations modulated by attention reflect visual experiences (for example, Cutrone et al 2014 as an early example); and quite a few examining how multiple levels of the visual hierarchy are modulated by attention (pretty much any visual attention paper using retinotopy). Similarly, lines 53-55 appear to go beyond the data in previewing the data – in what sense does top-down attention 'override' the bottom-up stimulus representation (rather than simply augmenting or enhance it). I would recommend toning down these claims.
2. Small number of participants. I appreciate that these experiments involve large scale data collection and so can only involve a small number of participants, but the limitation to five (of whom only a single participant was female) is a major issue. This is particularly the case given the heterogeneity of the key result (Fig 2c) across participants, where at least one participant and possibly two appears not to result in attended image classification greater than chance. The findings should perhaps not be presented as an average across five but for this critical result as a case-study plus replication. Similarly it should be made clearer that much of the data used in this study have been previously analysed and published.
3. Methodology description. It is very difficult for the reader to understand the link between the results described in the main text and the headings in the Methods, particularly as the latter do not correspond to the former. This is a particular difficult in understanding the critical contrast decoding analysis (whose results are presented in Figure 3) where the corresponding Methods are very challenging to locate and understand. The rationale for presenting each of the nineteen layers of the DNN separately is in particular very difficult to follow, and the biological significance of this in the context of figure 3 is very difficult to follow.
4. Indirect inferences. As the paper proceeds, the connection between the analyses and the underlying

biology becomes very hard to understand and the corresponding inferences more challenging. For example, by lines 139-144 the analyses presented appear to relate to the use of a decoded feature pattern (itself an indirect representation of brain activity) to be used to determine which of a small number (an artificially constrained problem) of image feature patterns the decoder was decoding. This accuracy is then used as a proxy for making inferences about the strength of attentional modulation in visual areas. But if I have understood correctly, the relationship of the decoded features to (for example) raw BOLD signal modulations is very indirect and does not necessarily reflect a simple amplitude modulation as there are other reasons why decoding accuracy and/or features might change in a multivariate set of features. Consequently I am not very comfortable with a simple interpretation of these findings in terms of amplitude modulation of signals in visual areas, as the paper seeks to do.

Overall this is a very interesting and clever method, but the nature of the biological insight provided is rather less clear which lowers overall interest in the significance of the findings for our understanding of how attention operates in the human brain.

Title: Attentionally modulated subjective images reconstructed from brain activity

We would like to thank the reviewers for their thoughtful comments and useful suggestions. We appreciate the constructive feedback, and we believe it has helped us clarify and improve the manuscript. We have revised our manuscript to address all the issues raised by the reviewers. Please find below detailed, point-by-point responses to the comments, and indications of where changes have been made in the manuscript.

Response to Reviewers

Reviewer #1:

“This paper demonstrates that it is possible to reconstruct images viewed by human subjects in an fMRI scanner in an ambiguous situation where two images are superimposed, and the subject is instructed to attend to one of them. Importantly, the reconstruction favors the attended image, demonstrating that the top-down modulation of neural signals by attention can be “read out” by fMRI decoding methods. This scientific conclusion, if not surprising (given prior knowledge from primate physiology), is important enough to merit publication. The reconstruction method and obtained images establish a new state-of-the-art for fMRI-based image reconstruction.

The paper is clearly written and structured, the arguments are sound and logical, and appropriate details are provided to allow replication of the results. In fact, the manuscript is in very good shape, possibly as a result of previous rounds of review in other journals (?). Furthermore, the decoding technique is noteworthy, drawing on state-of-the-art deep learning models. Finally, the novel dataset collected by the authors will be made available to the community on a data-sharing platform, and has the potential to become a standard dataset in the field. For all these reasons, I am enthusiastic about the manuscript, and I believe it can be published pretty much as it is.”

Response: We would like to express our deepest appreciation to the reviewer for endorsing the importance of our manuscript and would also like to thank the reviewer for recognizing the value of our data set. Following suggestions from the editor and other reviewers, we have collected data from two more subjects and made these datasets available from open data repositories. We believe that the potential utility of our datasets has been further enhanced by the confirmed replicability of the findings from these data.

Reviewer #2

“I thank the editor for the opportunity of reviewing this elegant study on the influence of top-down processes during the perception of stimuli. The topic is new, and it will add a significant contribution to the field of neural representations. However, few concerns must be addressed. A major issue concerns the effect size of the manuscript’s results. As a general recommendation, I would ask to author to report the effect size and the power of their main findings.”

Response: We would like to thank the reviewer for recognizing the novelty and significance of our manuscript. We also thank the reviewer for the constructive comments that helped us to improve our paper. We have revised our manuscript to address all the concerns raised by the reviewer. Specifically, given the reviewer’s concerns about the effect size and power of our analysis, we have tried to convince the reviewer of the strength of our results by using more conservative thresholds for statistical tests and power analysis (e.g., Figs. 3 and 4) and by showing effect sizes for all the main analyses (lines 116 and 121 for Fig. 2c; Figs. 3c and 5c and Supplementary Figures 3, 5, and 7). Furthermore, additional data from two new subjects showed results consistent with other subjects (e.g., successful reconstruction of attended images; Fig. 2c), supporting the replicability of our results. For a point-by-point response to the reviewer’s comments, please see below.

Comment #1

“1) In figure 3b, the top panel shows the correlation between the reconstructed image and the superimposed image. The graph shows that the peak correlation is around a low/medium effect size for most DNN layers for the first trial. Nevertheless, the correlations showed for the other two trials reach a small effect size only for few (highest) layers. These results indicate that only a few layers correlate with the weighted contrast. I would ask the authors to discuss how this result can be interpreted in terms of the degree of attentional modulation. I would also ask the author to statistically quantify the difference among the trials and modify the y axis to indicate the 0.1 correlation point (the least coefficient around which correlation can be assumed with a small effect size).”

Response: We apologize for the lack of clarity and thank the reviewers for giving us the opportunity to discuss how the strength of the attentional modulation should be interpreted in this analysis. As the reviewer pointed out, we admit that the mean correlation coefficients observed in Fig. 3 in the previous manuscript are

around 0.1, which is generally considered low. However, although the correlations obtained in the attention trials (Fig. 3 in the previous manuscript) were seemingly low, these correlations were comparable to those in the single-image trials (range of 0.05 to 0.2, Supplementary Figure 1a), which showed high image identification accuracy (>80% for most DNN layers, Supplementary Figure 1b) and image reconstructions of high enough quality to distinguish objects (84.7%, averaged across subjects, Fig. 2b, c, Supplementary Figure 1c). These results indicate that such a range of correlations (in the range of 0.05-0.2) is high enough to represent the identity of the images and is not negligible.

More importantly, as explained in the main text (lines 133–134), this analysis was performed to model the reconstructed images (or decoded features) from the attention trials using a superposition of attended and unattended images with weighted contrasts. Please note that the main focus of this analysis was the contrast of weighted superpositions that yields the highest correlation to a decoded feature pattern (peaks, circles on the lines in Fig. 3b), not the net correlations to the decoded features. Also note that even when a high correlation coefficient is observed, it does not indicate any effects of attention if the contrast showing the peak of the correlation is consistently around 50/50% (presented image), and if the correlation difference between the attended and unattended images (100/0% and 0/100%, respectively in Fig.3) is small (c.f., Fig. 4). Therefore, the net value of the correlation coefficient can be considered not crucial in assessing how attention biases the reconstructions and decoded features to the image attended by the subjects. Indeed, we see some examples of good reconstructions with correlation coefficients in the same range (around 0.01). So as not to mislead the readers, we have replaced examples in Fig. 3b with the same range of the y-axis as that of Figs. 3b bottom in the previous manuscript (note that we have largely changed Fig. 3 in the revised manuscript).

Taken the reviewer's comment into considerations, to make the main focus of this analysis clearer, we have recreated figures (Figs. 3c, d, and Supplementary Figure 3) to focus on shifts of peak contrasts that showed the highest correlations to decoded features. In Fig. 3c, we present density plots with mean shift sizes, confidence intervals, and individual data points (corresponding to each image pair). In Fig. 3d and Supplementary Figure 3, we present the effect sizes of the peak shifts for all combinations of brain areas and DNN layers. The analysis clearly showed significant biases of the peak positions with medium-to-large effect sizes often exceeding 0.5 on average and reaching 1.0–2.0 in individual subjects' results (Fig. 3d and Supplementary Figure 3), which were induced by attention and were almost comparable with the degree of attentional biases on behaviorally evaluated appearance (55.6%, Fig. 3c bottom). The results indicate robust attentional modulation across multiple-levels of visual areas and hierarchical visual features (DNN layers).

Regarding this point, we have revised our manuscript to quantitatively

evaluate the degree of the peak shifts and present variances of the shifts of peak contrasts by showing distributions and by computing confidence intervals [C.I.] of mean peak shifts estimated across pairs (Fig. 3c). We have also shown the effect size of those peak shifts in Fig. 3d and Supplementary Figure 3 while performing statistical tests to examine the significance of the peak shifts for all combinations of brain areas and DNN layers (lines 157–160 and lines 169–173). Additionally, to clarify the purpose of this analysis and avoid misunderstandings, we have added a new section for this analysis in Methods, detailing the motivation and procedures. Please see Methods: “Evaluation of attentional modulation by weighted image contrast” (lines 539–565) for this change. Moreover, we have added the interpretation about the seemingly low correlations in the feature decoding analysis in Supplementary Figure 1.

Regarding the reviewer’s request to statistically quantify the difference of correlation coefficients among the trials and that to modify the y-axis to show a correlation point of 0.1 (Fig. 3b bottom in the previous manuscript), we are sorry, but we have decided to change the visualization of this figure as denoted above. In the new figure, we did our best to clarify variances and effect sizes that are the most important concerns suggested by the reviewer. We hope these changes can resolve the reviewer’s concerns and provide more detailed information addressed in this analysis.

Comment #2

“2) In figure 3b, the bottom panel represents the correlation between the reconstructed image and the superimposed image, averaged for trials and participants. The graph reveals that only a few layers reach the 0.1 coefficient. As above, I would ask the authors to discuss only the correlations that display a small to high effect size.”

Response: We appreciate the reviewer’s suggestion and agree that it is important to interpret results with strict statistical criteria carefully. However, as noted above, here, the effect of the attentional modulation cannot be evaluated by the net values of the correlation coefficients but should be evaluated by the degree of the peak shift of the contrast that shows the highest correlation to the reconstructed image (or decoded feature pattern). Then, we have changed the figures of this analysis (Fig. 3 and Supplementary Figure 3) to directly evaluate the effect size of the shifts of the peak correlations, showing medium-to-large effect sizes (often exceeding 0.5 in Fig. 3d and reaching 1.0–2.0 in Supplementary Figure 3). Regarding the discussion of this issue, please also see the response to the first comment as well as the revisions in our manuscript (Fig. 3 and Supplementary Figures 1 and 3).

Comment #3

“3) Similarly, the results displayed in Figure 3c seem to contrast with the results discussed in the text. In fact, the higher layers seem to display a weak correlation with the weighted superimposition only in higher visual cortices. This result appears to be confirmed by the identification accuracy analysis, where lower visual cortices exhibit lower performances in attended image identification.”

Response: We would like to thank the reviewer for carefully reading our manuscript and checking the consistency between the text and figures. We assume that the reviewer’s comment is based on the idea that the correlations between a weighted superposition and a reconstruction (decoded feature pattern) should have a higher correlation coefficient. However, as we have mentioned in our responses to the first and second comments, in order to evaluate the effect of attentional modulation, the current analysis focused on how the contrast that shows the highest correlation to the decoded feature pattern changes from the contrast corresponding to the presented image (50/50%; Fig. 3a). Accordingly, we have recreated figures (Fig. 3c, d and Supplementary Figure 3) to focus on the peak shifts (please see the responses to the first and second comments).

Admittedly, as noted by the reviewer and discussed in our manuscript, V1 tended to be less accurate in identifying attended images than higher visual areas (Fig. 4b left). However, in Fig. 3d, features of most layers, especially around relatively lower-to-middle layers, decoded from V1 exhibited peak shifts toward attended images with medium-to-large effect sizes (0.5–1.0, see Supplementary Figure 3 for results of individual subjects). Furthermore, the significant peak shift toward the attended image was observed in multiple combinations of brain areas and DNN layers, consistent with our claim that attentional modulations were observed across visual areas and levels of hierarchical visual features (lines 173–175). Similar to these results, the results of the identification analysis also showed weak but significant accuracies even from V1 around middle DNN layers (Fig. 4). For a discussion of these points, see the responses to the first and second comments above. See also the revisions made for clarification (e.g., Fig. 3c, d, Supplementary Figures 1, and 3, and Methods: “Evaluation of attentional modulation by weighted image contrast” [lines 539–565]).

Comment #4

“4) Regarding the identification analysis, I would ask the authors to provide the power of their binomial and non-parametric analyses, given the low number of participants. Furthermore, the correction for multiple comparisons may be necessary for this analysis.”

Response: We appreciate the reviewer’s helpful suggestion to provide power and to correct statistical thresholds for multiple comparisons in the identification analysis. We agree that it is necessary to use power and tighter thresholds for reducing statistical errors, especially because the number of conditions tested in this analysis was large.

Following the suggestion by the reviewer, we re-analyzed the results of the identification analysis using stricter statistical criteria. First of all, as the previous analysis relied on the relatively large number of samples (trials) for statistical tests (e.g., ~720 for attention trials), which also could be biased due to the potential effect of image saliency (see lines 100–106), we first pooled identification results for each pair by averaging over all trials and attention conditions (a total of 16 trials from two attention conditions for each image pair) to yields a total of 45 data points for each image pair (see lines 184–187 and Methods: “Identification analysis” for the detailed procedure). We then performed t-tests to examine the significance of the mean identification accuracies for all combinations of brain areas and DNN layers with the correction for multiple comparisons ($p < 0.01$, Bonferroni correction by the numbers of brain areas and DNN layers; see the legend of Fig. 4). Furthermore, we also performed the post hoc power analysis and confirmed that all the significant results obtained by the analysis above showed powers higher than 0.8 (see the legend of Fig. 4). Moreover, we have added a new figure that shows the effect size of the identification accuracy by calculating Cohen’s d , which ranges around 0.5 to 2.0, to demonstrate the strength of our results (Supplementary Figure 5). We believe that these changes further strengthen the results of our manuscript for both type 1 and type 2 errors. Please see Fig. 4, Supplementary Figure 5, and Methods: “Statistics and reproducibility” (lines 633–651) in the revised manuscript for more details on these changes.

In addition to these changes, at the requests by the editor and another reviewer, data were collected from two additional subjects. Analyses using the data from these two subjects (Subject 6 and 7) yielded results consistent with the main claims of our manuscript (e.g., successful reconstructions of attended images in Fig. 2), confirming the replicability of our experiment and analysis. Related to this change, we revised our manuscript to show results of statistical tests for individual subjects to examine the replicability of our main findings (lines 113–122, 157–160, and 169–173). Furthermore, in response to the comment by another reviewer, we have performed an additional analysis to examine the effect of attention on amplitudes of individual DNN units in Fig. 5 and Supplementary Figures 6 and 7. Please see the comments and responses to the other reviewers for details of these changes.

“Minor points”:

Comment #5

“In the results (Figure 2c; Supplementary Figure 4) and the discussion sections, the authors state that subjects with higher accuracy performance in single-image do not perform better at attended image reconstructions. I wonder whether the authors performed a within-group statistical comparison of single vs. attended accuracy performance to support this claim.”

Response: Thank you for the comments. In fact, the statement was not based on a statistical comparison but merely pointed out qualitative differences in performance trends between the attention and single-image conditions. We have checked the correlation between the average identification accuracies of individual subjects for the attention and single-image conditions (Fig. 2c). However, the small number of subjects available for performing a group analysis made it difficult to obtain stable estimates of the correlation ($p > 0.10$, $n = 5$ or 7). Therefore, we decided not to draw any conclusions on this point but to indicate the necessity of future research in the discussion section (lines 270–279).

Comment #6

“Has the effect of the time gap (2 years) been controlled in the accuracy analysis?”

Response: We thank the reviewer for asking an important question. We recognize that the two-year time gap may negatively impact the accuracy of our analyses. Actually, we did not control for the effect of the gap. However, for some subjects (Subject 1–3), we confirmed that the quality of reconstructed images of single-image trials was qualitatively similar between the datasets collected over two years (data from Shen et al., 2019 and data from this study), indicating the long-term robustness of our experiments and analysis.

Comment #7

“The links on page 17, line 380, and page 19 lines 447-450, redirect to “not found” content.”

Response: Thank you for pointing this out. We also apologize for the incorrect hyperlinking of the URL; the URL itself is correct, but we found that the hyperlink was not working properly because the new line started in the middle of the URL.

We now carefully updated the hyperlinks to directly jump to the correct websites by clicking the URLs (lines 357, 396, 463, 516–517, 521, 654–657, and 660–661).

Reviewer #3

“This is an interesting manuscript that investigates how a neural network can decode and reconstruct images shown to people from recordings of their brain activity. The manuscript builds on earlier work by the same authors showing that such reconstructions from participants’ brain activity can be similar to the image presented to the participant, to now examine how such reconstructions are affected by the exercise of voluntary attention. The principal finding is that the exercise of voluntary attention leads to changes in the reconstructed image, leading the authors to infer that this reflects a change in the brain activity that serves as input to the decoder.

This is a reasonable conclusion, but emphasises the challenge this paper has in terms of novelty. Much is known about how attention modulates brain activity, and broadly speaking this paper confirms what is already known. The novelty of this paper therefore rests primarily in the application of these decoding methods to brain activity rather than in any specific biological insights.

I have a number of more specific suggestions for improving the paper: ”

Response: We would like to thank the reviewer for providing a list of helpful suggestions to improve our manuscript. We took these comments seriously and revised our manuscript to address the issues raised by the reviewer. Specifically, as the previous manuscript lacks the clarity of the differences and novelties from previous studies, we have revised the manuscript to make them clearer. Furthermore, we have added a new analysis to see how attention selectively modulates amplitudes of a particular set of decoded features by comparing amplitudes of decoded features in attention trials for different sets of DNN units that are relevant to attended, unattended, and the other images (Fig. 5 and Supplementary Figures 6 and 7). Additionally, in response to the suggestion by the editor and the reviewer, we have collected and analyzed data from two more subjects (one female and one male) and confirmed that the results were similar to those obtained from other subjects who showed reliable attentional modulations on reconstructed images. For point-by-point responses to the comments by the reviewer, please see below.

Comment #1

“1. Tone down claims in the Introduction. Lines 29-31 are in my view overstated – there are quite a few papers showing that neural representations modulated by attention reflect visual experiences (for example, Cutrone et al 2014 as an early example); and quite a few examining how multiple levels of the visual hierarchy are modulated by attention (pretty much any visual attention paper using retinotopy). Similarly, lines 53-55 appear to go beyond the data in previewing the data – in what sense does top-down attention ‘override’ the bottom-up stimulus representation (rather than simply augmenting or enhance it). I would recommend toning down these claims.”

Response: We apologize for the lack of clarity in our manuscript. We admit that the claims pointed out by the reviewer certainly need to be elaborated upon in order to convey our ideas more accurately. We will explain the details for each of the points raised by the reviewer below.

First, regarding the issue of novelty as an attention study (lines 29–31 in the previous manuscript), we recognize that Cutrone et al. (2014) investigated the relationship between visual experiences (appearance) and neural responses. They examined the effect of attention on appearance using simulated activities from simple contrast-response functions, showing that the change in input baselines could account for psychophysical results on perceived contrasts of Gabor stimuli (gratings within apertures) under attention. However, the study did not use “real” neural data but used simulated (or synthetic) neural activities generated by contrast-response functions. In addition, due to the simple nature of their model, the study was not able to investigate neural responses to complex natural images or the hierarchically organized visual features that span from low- to middle- to high-level visual features as modeled by deep neural networks.

Furthermore, Störmer et al. (2009) studied human EEG responses evoked by Gabor stimuli while spatial attention (left vs. right visual fields) was controlled by cross-modal cueing. They found that the enhancement of apparent contrast of the cued stimulus was accompanied by an enlarged EEG response in the contralateral hemisphere, demonstrating a link between appearance and neural activity under attentional modulation. Our study extends it to fine-grained and hierarchical neural representations of complex and naturalistic stimuli, enabling the reconstruction of subjective percepts.

Moreover, as suggested by the reviewer, many studies have reported the effects of attentional modulation across multiple levels of the visual hierarchy in the brain (e.g., V1–V4). However, those studies focused on specific types of visual features and categories (e.g., edge orientations, motion directions, semantic categories, and retinotopy) using experimentally designed stimuli to examine the effects of attention on neural representations. In contrast, our approach is based on hierarchical DNN features that are discovered via the training with a massive

dataset of natural images and thus are difficult to be designed by an experimenter. It enables us to examine millions of naturalistic visual features (not the multiple levels of visual cortical areas, but the multiple levels of visual “features”), many of which are relevant to neural representations in the human brain (see the sixth paragraph of the Discussion section of our manuscript [lines 306–319]). This approach reveals that attentional modulations are found in a broad range of hierarchical representations (in both visual hierarchical areas and hierarchical DNN features) constrained by the correspondence between brain areas and DNN layers.

To summarize our response to the first point of the reviewer’s comment, by using the hierarchical features of DNNs, our study extends previous findings on the relationship between visual experiences (appearance) and neural responses, tested in simple settings, to more complex situations (natural images) and shows robust attentional modulation across visual areas and multiple levels of hierarchical visual features. Considering the reviewer’s comments, we have revised the Introduction section to make these points clearer and also revised the Discussion section to detail the discussion above. Please see the second paragraph in the Introduction (lines 28–34) and the seventh to ninth paragraphs in the Discussion (lines 306–339) for these changes.

As for the later point of the reviewer’s comment (“*lines 53–55, in what sense does top-down attention ‘override’ bottom-up stimulus representation*”), we originally intended to indicate the finding that correlations to decoded feature patterns can be higher with image feature patterns from attended images (top-down) rather than those from presented images (bottom-up), as demonstrated in Fig. 3b. However, as we agree that our statements in the Introduction section were not enough to describe these points, we revised our manuscript while toning down the claim according to the reviewer’s suggestion (line 55).

Comment #2

“2. Small number of participants. I appreciate that these experiments involve large scale data collection and so can only involve a small number of participants, but the limitation to five (of whom only a single participant was female) is a major issue. This is particularly the case given the heterogeneity of the key result (Fig 2c) across participants, where at least one participant and possibly two appears not to result in attended image classification greater than chance. The findings should perhaps not be presented as an average across five but for this critical result as a case-study plus replication. Similarly it should be made clearer that much of the data used in this study have been previously analysed and published.”

Response: We appreciate the reviewer for pointing out the important issue of our study. We agree that the results varied from subject to subject and that the performance of one subject in the attended image reconstruction analysis was close to the chance (Fig. 2c left). We took this point seriously and decided to follow the suggestions by the editor and the reviewer to collect additional data from two more subjects. The same experiments and analyses were performed on the two additional subjects (one female and one male; Subject 6 and 7), and the results were comparable or better than other subjects who showed moderate reconstruction performance (e.g., Subject 1–3). Furthermore, following the suggestion by the reviewer, we present our results as a case study plus replication by performing statistical testing for individual subjects first with each of the original five subjects (only Subject 5 failed to show the significant result in Fig. 2c) and then with additional two subjects (see lines 113–122 for the evaluations of reconstructions).

In summary, these results from the additional subjects confirmed the replicability of our experiments and analyses, improving the generalizability of our findings. In light of these results, we have revised our manuscript and re-estimated the summary performances of all our analyses. Please see Figs. 2–4, Supplementary Figures 1–5, and the summary performances in the main text (lines 113–122, 157–160, and 169–173). Please note that in response to the comments by another reviewer, we have revised Fig. 3 to focus on the shift of the peak contrast that showed the highest correlations to decoded features, in which we presented density plots, effect sizes of the shift size, and their confidence intervals and performed statistical tests on these results.

Regarding the latter part of this comment, we apologize for lacking the information that the training data of Subject 1–3 were from the previous study (Shen et al., 2019) from the main text. Although we have described this information in the Methods: “Subjects” and “Experimental design,” we have also added the same information in the main text to clarify this point. Please see the first paragraph of the Results section (lines 60–74) for this change.

Comment #3

“3. Methodology description. It is very difficult for the reader to understand the link between the results described in the main text and the headings in the Methods, particularly as the latter do not correspond to the former. This is a particular difficult in understanding the critical contrast decoding analysis (whose results are presented in Figure 3) where the corresponding Methods are very challenging to locate and understand. The rationale for presenting each of the nineteen layers of the DNN separately is in particular very difficult to follow, and the biological significance of this in the context

of figure 3 is very difficult to follow.”

Response: We apologize for the lack of clarity and would like to thank the reviewer for the opportunity to improve the readability of our manuscript. We admit that it was difficult to find the link between the result and methods in the previous manuscript. In response to the reviewer’s comment, we have revised our manuscript to clarify links between results and methods by inserting references to the Methods section for each analysis (e.g., “Feature decoding analysis,” “Visual image reconstruction analysis,” “Evaluation of reconstruction quality,” and so on). Furthermore, according to the comment by the reviewer, we have added a new section that primarily describes the method of the contrast analysis (Fig. 3 and Supplementary Figures 3 and 4) in the Method section (Methods: “Evaluation of attentional modulation by weighted image contrast”, lines 539–565). Additionally, taking the comments by reviewers into account, we have revised figures of the contrast analysis (Fig. 3c and d and Supplementary Figure 3) to enhance the clarity of the main focus of this analysis (the shift of peak contrast). Moreover, regarding the rationale for presenting each of the nineteen layers of the DNN separately in Fig. 3, we have added explanations in both the main text (lines 28–34, and 138–140) and Methods “Evaluation of attentional modulation by weighted image contrast” (lines 539–565). In this analysis, we decided to show results from individual layers because one of the purposes of this study is to investigate the effect of attention on naturalistic visual features and their neural population-level representations, both of which are thought to have hierarchical organizations (see the second paragraph of the Introduction section [lines 28–34] and the sixth paragraph in Discussion section [lines 306–319]). Please see the new method section (Methods: “Evaluation of attentional modulation by weighted image contrast,” lines 539–565) for details. We hope these changes would contribute to enhancing the clarity and readability of our manuscript.

Comment #4

“4. Indirect inferences. As the paper proceeds, the connection between the analyses and the underlying biology becomes very hard to understand and the corresponding inferences more challenging. For example, by lines 139-144 the analyses presented appear to relate to the use of a decoded feature pattern (itself an indirect representation of brain activity) to be used to determine which of a small number (an artificially constrained problem) of image feature patterns the decoder was decoding. This accuracy is then used as a proxy for making inferences about the strength of attentional modulation in visual areas. But if I have understood correctly, the relationship of the decoded features to (for example) raw BOLD signal modulations is very indirect and does not necessarily reflect a simple

amplitude modulation as there are other reasons why decoding accuracy and/or features might change in a multivariate set of features. Consequently I am not very comfortable with a simple interpretation of these findings in terms of amplitude modulation of signals in visual areas, as the paper seeks to do.”

Response: We appreciate the reviewer’s comment for the concern about the connection between the analyses and the underlying biology. We agree that it is important to confirm whether the methodology used in this study can be justified to make correct inferences about the information represented in the brain (in this case, attentional modulation). In the following, we first provide the rationale of the validity of our decoding approach and then explain the results of the new analysis performed for clarifying the potential mechanism that links between modulations of BOLD signals and decoding results.

First of all, we apologize for our failure to clearly communicate our ideas and misleading the reviewer about the rationale of our decoding approach. Please note, however, that we do not intend to equate the modulations of raw BOLD signals with the modulations of decoding accuracy and/or features. On the contrary, we assume that the modulation of the raw BOLD signal is not a reliable measure of the information represented in brain activity because of its low signal-to-noise ratio. Instead, we consider that outputs of multivariate decoding models represent the information in the target features more accurately than the raw BOLD signals, as shown in many previous studies (e.g., Haxby et al., 2002; Kamitani & Tong, 2005; Hamilton & Tong, 2010). In fact, our approach, which consists of multi-level DNN feature decoding analysis and the deep image reconstruction method, has previously been shown to be useful for reliably reconstructing (decoding) perceived and imagined images from brain activity patterns (Horikawa & Kamitani, 2017; Shen et al., 2019). Furthermore, in the present study, we confirmed that decoded features from the trained models were useful to accurately identify and reconstruct images even from single-trial brain activity patterns (Supplementary Figure 1). To clarify the rationale of our decoding approach, we have first enhanced the discussion in the legend of Supplementary Figure 1.

Admittedly, as the reviewer pointed out, the identification analysis via decoded features (Fig. 4, lines 139-144 in the previous manuscript) was performed in an artificially constrained setting (i.e., in a pair-wise manner). In addition, because this analysis was performed using a multivariate set of features (using correlations between feature patterns), it is important to exclude potential confounding factors to correctly attribute the results to the effect of attention (as discussed in Naselaris & Kay, 2015). Unlike conventional classification methods, our decoders were specifically tuned (trained) to predict the responses of individual DNN units (i.e., explicit models of representations as in Naselaris & Kay,

2015), which could in part resolve the ambiguity of the source of the identification accuracies by attributing to the variations in feature representations. Thus, contrary to the reviewer's concern, our feature decoding approach is robust to potential confounding factors often accompanied in multivariate analyses. We have also revised our manuscript to explain this point in the Discussion section (lines 295–305).

To directly confirm this point, we have performed an additional analysis to examine how attention affects the decoded features of individual DNN units (Fig. 5). In this analysis, we evaluated the amplitude modulation of the decoded features in attention trials for different sets of DNN units, each of which was defined for the attended image, unattended image, and the other images for each attention trial (see Methods: "Evaluation of amplitude modulations on decoded features" for the detailed procedure). Noticeably, the analysis showed greater amplitudes for the attended feature set than the unattended feature set and the feature sets for the other images, indicating that top-down attention is reflected in the amplitudes of a particular set of features relevant to the attended image (Figs. 5b, c, and Supplementary Figures 6 and 7). This may arise from amplitude modulations of neural feature representations because linear transformations link brain activity and decoded feature values. These results suggest a potential mechanism based on selective amplitude modulations of neural feature representations, which may underlie the effect of attention on decoding accuracy (reconstructions and decoded features).

In summary, we seek to evaluate the effect of attentional modulation concerning the multi-level visual information represented in patterns of brain activity rather than the amplitude of raw BOLD signals. Such multi-level visual information can be reliably detected via decoding of hierarchically organized DNN features as demonstrated in previous studies (Horikawa & Kamitani, 2017; Shen et al., 2019) and our validation analysis (Supplementary Figure 1). In our analysis, the decoders were specifically trained to predict feature values of individual DNN units, eliminating potential confounding factors that may obscure the interpretations of decoding accuracy. Indeed, the additional analysis demonstrated that attention selectively modulates the amplitudes of decoded features for a particular set of DNN units relevant to attended images (Fig. 5 and Supplementary Figures 6 and 7). Thus, we believe that decoding accuracy/features can be used as reliable measures of attentional modulation associated with hierarchical visual features encoded in the brain.

Regarding the concern raised by the reviewer, we have revised our manuscript to clarify these points. Please see the second paragraph of the Introduction section (lines 28–34; also see the response to the second comment by the reviewer relevant to this revision) and the fifth paragraph of the Discussion section (lines 295–305). Furthermore, please see Fig. 5 and Supplementary Figures 6 and 7 for the results of additional analysis of amplitude modulations on

decoded features. We hope that the reviewer's concerns about the problem of the indirect inference have been addressed by these revisions.

Comment #5

"Overall this is a very interesting and clever method, but the nature of the biological insight provided is rather less clear which lowers overall interest in the significance of the findings for our understanding of how attention operates in the human brain."

Response: We sincerely thank the reviewer for the positive comments on our approach and thoughtful suggestions to improve the manuscript. In response to the feedback we received, we have revised the manuscript and explained our views to address all the issues raised. We believe that these changes have enhanced the clarity regarding the significance of our manuscript.

In the following, we show the revised figures and legends relevant to the revisions above according to the instruction of the journal.

Fig. 2 Reconstructions from individual attention and single-image trials. **a** Reconstructions from attention trials. Reconstructions with relatively high rating accuracies are shown (see Supplementary Figure 2 for more examples; see Methods: “Evaluation of reconstruction quality”). For each specific presented image, two reconstructions from the same subjects are shown for trials with different attention targets. **b** Reconstructions from single-image trials. Images with black and gray frames indicate presented and reconstructed images, respectively (see Supplementary Figure 1c for more examples). **c** Identification accuracy based on behavioral evaluations. Dots indicate mean accuracies of pair-wise identification evaluations averaged across samples for each paired comparison (chance level, 50%; see Methods: “Evaluation of reconstruction quality”). Black and red lines indicate mean and lower/upper bounds of 95% C.I. across pairs. **d** Scatter plot of attended and single-image identification accuracy based on behavioral evaluations. Dots indicate mean accuracies averaged over samples for each paired comparison and subjects. The red line indicates the best linear fit.

Fig. 3 Attentional modulation modeled by image contrast. **a** Evaluation procedure by weighted image contrasts. Correlations were calculated between decoded feature patterns and feature patterns computed from superpositions with weighted contrasts (5% steps; presented stimuli correspond to 50% contrast; normalized by the values in training images for each unit; see Methods: “Evaluation of attentional modulation by weighted image contrast”). **b** Correlation curves as a function of contrast for individual reconstructions with relatively high rating accuracies (decoded from VC). Circles indicate the contrasts with the highest correlations for each of 19 DNN layers denoted by colors. **c** Distributions of peak contrasts for each DNN layer (decoded from VC; averaged across subjects). The peak contrasts for 45 image pairs (cross markers) are shown with density plots (horizontal bars, 95% C.I.). The gray distribution at the bottom indicates contrasts of visual appearance evaluated in an independent behavioral experiment (averaged across subjects; see Methods: “Evaluation of visual appearance”). **d** Peak shifts at visual subareas (averaged across subjects). The effect sizes (Cohen’s d) of the peak shifts (difference from 50%) are shown with 95% C.I. for each area and layer. See Supplementary Figure 3 for the results of individual subjects.

Fig. 4 Identification by feature correlation. **a** Identification procedure. Pair-wise identification of an attended image or a single presented image was performed via decoded features obtained from each trial (chance level, 50%; normalized by the values in training images for each unit; see Methods: “Identification analysis”). **b** Identification accuracy for attention and single-image reconstructions (averaged across subjects). Mean identification accuracies are shown for all combinations of individual visual subareas and DNN layers. Statistical analyses were performed with the identification accuracies across 45 image pairs. Colored lines beneath data indicate the statistical significance of the difference from the 50% chance level (one-sided t -test, $p < 0.01$, Bonferroni correction by the numbers of brain areas and DNN layers), in which the power estimated by a post-hoc analysis was higher than 0.8 for all those significant results. See Supplementary Figures 4 and 5 for the results of individual subjects with effect sizes.

Fig. 5 Evaluation of amplitude modulations on decoded features. **a** Evaluation procedure by amplitude modulations. The amplitude of decoded features in each attention trial was evaluated for different sets of DNN features (“feature set” for the attended image, the unattended image, and the other images; normalized by the values in training images for each unit; see Methods: “Evaluation of amplitude modulations on decoded features”). **b** Amplitude of decoded features in attention trials (averaged across subjects). Mean amplitude averaged across units, trials, and subjects for the three types of feature sets are shown (decoded from VC; see Supplementary Figure 6 for the results of individual subjects). Each dot indicates the mean amplitude of each image pair. Black and red lines indicate mean and lower/upper bounds of 95% C.I. across pairs. **c** Cohen’s d of amplitude differences between the attended and the unattended feature sets at visual subareas (averaged across subjects). The effect sizes (Cohen’s d) of the amplitude differences are shown for each area and layer. Statistical analyses were performed with the Cohen’s d across 45 image pairs. Colored lines beneath data indicate the statistical significance of the difference (one-sided t -test, $p < 0.01$, Bonferroni correction by the numbers of brain areas and DNN layers). See Supplementary Figure 7 for the results of individual subjects.

Supplementary Figure 1 Feature decoding and reconstruction results for single-image trials. **a** Feature decoding accuracy for single-image trials. A decoding accuracy was evaluated by calculating a Pearson correlation coefficient between a pattern of decoded feature values and a pattern of image feature values computed from presented images for each sample (decoded from the visual cortex [VC]; normalized by the values in training images for each unit). Correlation coefficients were averaged across samples from the single-image trials (a total of 160 trials for each subject, colored dots), and the mean correlations averaged across subjects (gray bars) are shown for each layer of the VGG19 model. For all subjects and DNN layers, the feature decoding accuracies were significantly higher than the chance (one-sided t -test, $p < 0.01$, Bonferroni correction by the numbers of DNN layers and subjects). **b** Pair-wise image identification accuracy for single-image trials. Identification accuracy obtained by the pair-wise identification analysis is shown for each layer of the VGG19 model (decoded from VC; normalized by the values in training images for each unit; chance level, 50%; see Methods: "Identification analysis"). In the analysis, correlation coefficients were calculated between a pattern of decoded features and patterns of image features of two candidate images (one for true [presented], and the other for false), and the image with a higher correlation coefficient was selected as the predicted image. For each sample, pair-wise identification was performed for all pairs between one true image and the other nine false images used in the test session (nine pairs for each sample). The accuracy of each sample was defined by the proportions of correct identification. For all subjects and DNN layers, the identification accuracies were significantly higher than the chance (one-sided t -test, $p < 0.01$, Bonferroni correction by the numbers of DNN layers and subjects). Conventions are the same with Supplementary Figure 1a. **c** Examples of reconstructed images from single-image trials. The reconstructed images produced from samples of each of the single-image trials are shown for seven subjects (decoded from VC). Conventions are the same with Fig. 2b. It is noteworthy that even though the average correlation coefficient of feature decoding accuracy was not high (in the range of 0.05 to 0.2, Supplementary Figure 1a), the identification accuracy of the viewed images reached more than 80% for most DNN layers, and the quality of the reconstructed images reached a level sufficient to identify objects (84.7%, averaged across subjects; Fig. 2c). These high performances of the identification and reconstruction analyses indicate that correlations around such a range are not negligible. Furthermore, when we took a closer look at the relationship between the reconstruction accuracy from single-image trials (Fig. 2c right) and the feature decoding accuracy (Supplementary Figure 1a and b), we found that a group of subjects with relatively high decoding accuracy (e.g., Subject 3, 4, 6, and 7) performed better in the reconstruction analysis of seen images, and vice versa (see Subject 1, 2, and 5 for the case with relatively low accuracy). This positive relationship between performances of feature decoding and image reconstruction may confirm that decoded features with high accuracy can be interpreted as a reliable measure of the visual information represented in the brain, and it can be inferred that the decoding accuracy/features obtained from the attention condition reflect visual information modulated by attention.

Supplementary Figure 2 Examples of reconstructed images for attention trials. **a** Examples of attended image reconstructions with high behavioral rating accuracies. Reconstructed images with relatively high rating accuracies (higher than 80%) are shown. Conventions are the same with Fig. 2a. **b** Examples of attended image reconstructions with low rating accuracies. Reconstructed images with relatively low rating accuracies (lower than 60%) are shown. Failures of attended image reconstructions were categorized into clutter images, mixtures of two superimposed images, or images more similar to unattended images. **c** Reconstructed images from samples for trials without button responses. Reconstructed images obtained from samples for miss trials, in which subjects missed a button-press to indicate correct recognition of target images, are shown. The reconstructions from these miss trials tended to be not similar to either of the two superimposed images. **d** Reconstructed images from samples for trials with incorrect button responses. Reconstructed images obtained from samples for error trials, in which subjects incorrectly pressed a button to indicate target images, are shown. The reconstructions from these error trials sometimes produced images judged to be similar to non-target (or instructed to be unattended) images.

Supplementary Figure 3 Peak shifts at visual subareas for individual subjects. Conventions are the same with Fig. 3d. Subjects whose reconstructions from attention trials were evaluated highly (e.g., Subject 1–3 and 7; cf., Fig. 2c) tended to specifically show greater biases in decoded feature patterns. The results showed that the significant effect of attentional modulations observed in four of the initial five subjects (Subject 1–4) were replicated with the two additionally collected subjects (Subject 6 and 7).

Supplementary Figure 4 Identification accuracy for individual subjects. Mean identification accuracies of individual subjects are shown for all combinations of individual visual subareas and DNN layers. Conventions are the same with Fig. 4b. The results showed relatively greater variability among subjects in the accuracies of the attended image identification than those of the single-image identification, possibly due to the individual differences in the ability to direct their selective attention. Differences in brain areas that showed high attended image identification accuracies might be attributable to differences in their strategies for attention, as we did not explicitly provide specific strategies for their attempt of attention.

Supplementary Figure 5 Cohen's d of identification accuracy for individual subjects.

For each combination of brain areas and DNN layers, the Cohen's d of the identification accuracy was calculated by first subtracting 50 (chance level) from the identification accuracy averaged for each pair (45 pairs) and then by normalizing the accuracy by the standard deviation across pairs. Colored lines underneath the data indicate the results of statistical tests that are the same with those shown in Fig. 4b and Supplementary Figure 4. Conventions are the same with Supplementary Figure 4. On average, the results showed medium-to-large effect sizes in the attention condition, indicating the robustness of the attentional modulations on decoded features.

Supplementary Figure 6 Amplitude of decoded features in attention trials for individual subjects. Mean amplitudes averaged across units, trials, and image pairs for each subject are shown for the three types of feature sets (decoded from VC). Dots and bars indicate mean amplitudes for individual subjects and their average, respectively.

Supplementary Figure 7 Cohen's d of amplitude differences between the attended and unattended feature sets at visual subareas for individual subjects. The effect sizes (Cohen's d) of amplitude differences between the feature sets of attended and unattended images of individual subjects are shown for all combinations of areas and layers. Conventions are the same with Fig. 5c.

References

1. Cutrone, E. K., Heeger, D. J. & Carrasco, M. Attention enhances contrast appearance via increased input baseline of neural responses. *J. Vis.* **14**, 16 (2014).
2. Störmer, V. S., McDonald, J. J. & Hillyard, S. A. Cross-modal cueing of attention alters appearance and early cortical processing of visual stimuli. *Proc. Natl. Acad. Sci. U.S.A.* **106**, 22456–22461 (2009).
3. Klein, B. P., Harvey, B. M., & Dumoulin, S. O. Attraction of Position Preference by Spatial Attention throughout Human Visual Cortex. *Neuron* **84**, 227–237 (2014).
4. Naselaris, T., & Kay, K. N. Resolving Ambiguities of MVPA Using Explicit Models of Representation. *Trends Cogn. Sci.* **19**, 551–554 (2015).
5. Haxby, J. V., Gobbini, M. I., Furey, M. L., Ishai, A., Schouten, J. L., & Pietrini, P. Distributed and overlapping representations of faces and objects in ventral temporal cortex. *Science* **293**, 2425–2430 (2001).
6. Kamitani, Y., & Tong, F. Decoding the visual and subjective contents of the human brain. *Nat. Neurosci.* **8**, 679–685 (2005).
7. Harrison, S. A., & Tong, F. Decoding reveals the contents of visual working memory in early visual areas. *Nature* **458**, 632–635 (2009).
8. Horikawa, T. & Kamitani, Y. Generic decoding of seen and imagined objects using hierarchical visual features. *Nat. Commun.* **8**, 15037 (2017).
9. Shen, G., Horikawa, T., Majima, K. & Kamitani, Y. Deep image reconstruction from human brain activity. *PLOS Comput. Biol.* **15**, e1006633 (2019).

REVIEWERS' COMMENTS:

Reviewer #1 (Remarks to the Author):

I was very positive about this manuscript's initial submission. The latest version includes 2 additional subjects, making the dataset more useful for the community, and the conclusions stronger. I remain supportive, and believe that this study deserves to be published.

Reviewer #2 (Remarks to the Author):

I would like to thank the authors for considering the point raised and for providing additional data. The manuscript has much improved in terms of clarity. I am happy to recommend acceptance.

Reviewer #3 (Remarks to the Author):

I thank the authors for their consideration of my comments, their comprehensive response and the extensively revised manuscript. The revision is detailed and thorough, and the collection of data from two additional participants has significantly increased confidence in the findings. The textual revisions have also improved clarity substantially and made the paper - while complex - easier to follow for the uninitiated. I have no further comments or concerns to address.

Title: Attention modulates neural representation to render reconstructions according to subjective appearance

We would like to thank all the reviewers for their positive comments on the manuscript. Our manuscript has been improved through communications with the reviewers. We greatly appreciate their thoughtful comments and constructive feedback.

Reviewers' comments

Reviewer #1:

"I was very positive about this manuscript's initial submission. The latest version includes 2 additional subjects, making the dataset more useful for the community, and the conclusions stronger. I remain supportive, and believe that this study deserves to be published."

Reviewer #2

"I would like to thank the authors for considering the point raised and for providing additional data. The manuscript has much improved in terms of clarity. I am happy to recommend acceptance."

Reviewer #3

"I thank the authors for their consideration of my comments, their comprehensive response and the extensively revised manuscript. The revision is detailed and thorough, and the collection of data from two additional participants has significantly increased confidence in the findings. The textual revisions have also improved clarity substantially and made the paper - while complex - easier to follow for the uninitiated. I have no further comments or concerns to address."